# Ada-KV: Optimizing KV Cache Eviction by Adaptive Budget Allocation for Efficient LLM Inference

**Yuan Feng**[1,3†] **Junlin Lv**[1,3†] **Yukun Cao**[1,3] **Xike Xie**[2,3*] **S. Kevin Zhou**[2,3]

[1]School of Computer Science, University of Science and Technology of China (USTC)
[2]School of Biomedical Engineering, USTC
[3]Data Darkness Lab, MIRACLE Center, Suzhou Institute for Advanced Research

## Abstract

Large Language Models have excelled in various domains but face efficiency challenges due to the growing Key-Value (KV) cache required for long-sequence inference. Recent efforts aim to reduce KV cache size by evicting vast non-critical cache elements during runtime while preserving generation quality. However, these methods typically allocate compression budgets uniformly across all attention heads, ignoring the unique attention patterns of each head. In this paper, we establish a theoretical loss upper bound between pre- and post-eviction attention output, explaining the optimization target of prior cache eviction methods, while guiding the optimization of adaptive budget allocation. Base on this, we propose *Ada-KV*, the first head-wise adaptive budget allocation strategy. It offers plug-and-play benefits, enabling seamless integration with prior cache eviction methods. Extensive evaluations on 13 datasets from Ruler and 16 datasets from LongBench, all conducted under both question-aware and question-agnostic scenarios, demonstrate substantial quality improvements over existing methods. Our code is available at `https://github.com/FFY0/AdaKV`.

## 1 Introduction

Autoregressive Large Language Models (LLMs) have achieved significant success and are widely utilized across diverse natural language processing applications, including dialogue systems [1], document summarization [2], and code generation [3]. The widespread deployments of LLMs have propelled the development of their capacities to process extended sequences. For instance, GPT supports sequences up to 128K [4], Claude3 up to 200K [5], and Gemini-Pro-1.5 [6] up to 2M tokens. However, this growth in token length introduces significant challenges, particularly the rapid expansion of cache size during inference. For an 8B LLM, handling a single sequence of 2M tokens can require up to 256GB of cache, severely impacting both GPU memory efficiency and computational runtime efficiency.

In particular, the inference process, for each multi-head self-attention layer, consists of two phases: *prefilling* and *decoding*. In the prefilling phase, LLMs compute and store all Key-Value (KV) cache elements for the tokens in the input prompt. In the decoding phase, the model autoregressively uses the most recently generated token to retrieve information from the stored cache, producing the next output iteratively. The large KV cache size introduces two important efficiency challenges: First, it severely affects GPU memory efficiency, making it increasingly difficult to scale with longer inputs. Second, during decoding, increased I/O latency duo to cache access results in substantial delays, degrading runtime efficiency.

---

[†]Equal Contribution, [*]Corresponding Author: Xike Xie(xkxie@ustc.edu.cn)

39th Conference on Neural Information Processing Systems (NeurIPS 2025).

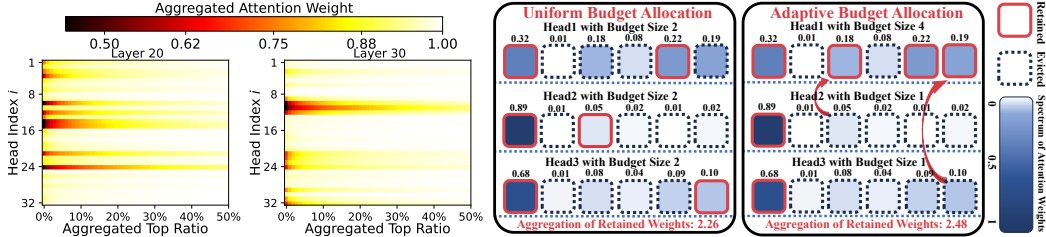

(a) Varied Attention Concentration Across Heads.   (b) From Uniform to Adaptive Budget Allocation

Figure 1: Adaptive budget allocation accommodates varying attention concentration across heads. *Left: Analysis using Llama-3.1-8B-Instruct shows most heads retain nearly all attention weights with a small cache (e.g., top 5%), while dispersed heads require larger cache proportions. Right: Adaptive allocation, which shifts budgets from sparse to dispersed heads, increases the aggregated retained attention weights (from 2.26 to 2.48) and reduces eviction loss compared to uniform allocation.*

To address the challenges posed by large KV cache sizes, various cache eviction methods have been developed [7, 8, 9, 10, 11]. These methods constrain the cache size to a predefined budget by retaining only a subset of elements in each attention head and evicting the rest. They reduces memory usage and accelerate decoding, facilitating efficient long-sequence inference. Most of these methods rely on a Top-$k$ selection based on attention weights, to identify and prioritize critical cache elements, deciding which to retain and which to evict.

However, existing methods uniformly allocate cache budgets across attention failing to account for the unique characteristics of each head. As shown in Figure 1a, attention heads exhibit diverse concentration patterns: some focus narrowly (or sparsely) on a small portion of the cache, while others distribute their attention more broadly [1]. This uniform allocation results in inefficiencies—either wasting cache budgets on heads with sparse concentration or incurring significant eviction losses in heads with dispersed distribution. Such imbalances degrade the trade-off between overall budget utilization and the quality of post-eviction generation. To address this issue, we analyze the impact of existing Top-$k$ eviction methods on attention outputs and propose the first adaptive budget allocation strategy, called *Ada-KV*, as illustrated in Figure 1b. This strategy dynamically reallocates budgets from heads with sparse concentration to those with more dispersed attention, improving the quality of post-eviction generation.

Our study begins by revisiting Top-$k$ eviction methods, demonstrating that, under specific budget allocation, these methods are equivalent to minimizing an upper bound of eviction loss between pre- and post-eviction attention outputs[2]. Based on this, we propose Ada-KV, a simple yet effective adaptive allocation strategy designed to integrate and enhance existing Top-$k$ eviction methods in a plug-and-play manner. Guided by minimizing the theoretical upper bound of eviction loss, Ada-KV adapts to the varying concentration patterns across attention heads, significantly reducing practical eviction loss.

By integrating the Ada-KV into two state-of-the-art (SOTA) methods, SnapKV [11] and Pyramid [9, 10], we have developed two integration cases: *Ada-SnapKV* and *Ada-Pyramid* [3]. These methods are evaluated using two comprehensive benchmarks, Ruler and LongBench, which include 13 and 16 datasets, respectively. Experimental results demonstrate that Ada-KV could effectively improves performance across various budget settings under the widely studied **question-aware** compression scenario [11, 10, 8]. Furthermore, in the more challenging **question-agnostic** [12] scenario—where compression is applied without leveraging question-specific information, an important scenario that has been largely overlooked—Ada-KV demonstrates even greater advantages. The main contributions are summarized as follows.

- **Adaptive Budget Allocation.** We identify a critical limitation in current KV cache eviction methods: their uniform budget allocation overlooks the unique attention patterns of individ-

---

[1]More visualizations can be found in Appendix A.11

[2]For simplicity, we use $L_1$ distance to quantify eviction loss; extensions to $L_p$ norms are possible but orthogonal to this work.

[3]The plug-and-play nature of Ada-KV enables broad applicability to existing methods and future developments, extending well beyond the two cases presented in this paper. Section 5 provides further examples of its adoption in subsequent work.

ual heads. To address this, we propose Ada-KV, the first adaptive budget allocation strategy, which enhances budget utilization across individual heads, leading to more efficient cache eviction methods.

- **Theoretical Insights.** We establish a theoretical framework for cache eviction by defining eviction loss and deriving its upper bound. This framework not only explains the optimization target of prior methods but also guides the design of Ada-KV, enabling principled and adaptive budget allocation.

- **Empirical Advances.** With efficient CUDA kernel implementations, Ada-KV offers plug-and-play compatibility, enabling seamless adoption and performance enhancements in existing methods. We evaluate Ada-KV by integrating it into SOTA cache eviction methods, and demonstrate significant improvements across 29 datasets from Ruler and LongBench, all under both question-aware and question-agnostic scenarios.

## 2 Related Works

### 2.1 Cache Eviction Methods

In the long-sequence inference, the vast scale of the KV cache elements leads to a memory-bound situation, causing significant memory burden and I/O latency [13]. Numerous studies have sought to mitigate by reducing the cache size, notably through the eviction of non-critical cache elements[4]. These methods are primarily divided into two categories: *sliding window eviction* and *Top-k eviction* methods. The sliding window eviction methods [14, 15, 16], exemplified by *StreamingLLM* [16], simply retain several initial cache elements and those within a sliding window, while evicting others. However, the undiscriminating sliding eviction of cache elements results in a significant reduction in generation quality. In contrast, Top-k eviction methods [7, 17, 18, 8, 9, 10, 11] identify and retain a selected set of $k$ critical cache elements based on attention weights, for enhancing the post-eviction generation quality. The adaptive budget allocation presented in this paper, tailored for Top-$k$ Eviction Methods, enhances post-eviction generation quality within the same overall budget.

In the study of Top-$k$ eviction methods, early work like FastGen [7] searches and combines multiple strategies, such as maintaining caches of special elements, punctuation elements, recent elements, and Top-$k$ selected elements, based on the characteristics of attention heads. H2O, as the representation [8, 17, 18], develops a Top-$k$ based eviction scheme that leverages the query states of all tokens to identify critical cache elements. However, under unidirectional mask in LLM computations, aggregating attention weights across all query states often causes recent KV cache elements to be mistakenly evicted, degrading the quality of subsequent generations. Recent works, such as SnapKV [11] address this issue by using query states within an observation window to identify critical elements, thereby achieving SOTA performance. Subsequent methods, such as Pyramid [9, 10], further introduce budget allocation across different layers. However, existing Top-$k$ eviction methods typically assign the overall budget uniformly across different heads. In contrast, Ada-KV, which has demonstrated superior theoretical and empirical results through adaptive budget allocation, provides a novel strategy to optimizing these methods.

### 2.2 Sparse Attention Methods

Sparse attention methods are conceptually related to KV cache eviction, but differ fundamentally in their approach [19, 20, 21, 22, 23, 24]. The key difference is that KV cache eviction retains only a subset of the KV cache, while sparse attention methods keep all entries but selectively utilize only a critical subset during computation. Consequently, sparse attention methods do not reduce the memory footprint of the KV cache, thus often necessitating cache offloading to CPU memory [21, 22, 23] or even disk storage [24]. Additionally, the two technique lines are, in fact, orthogonal. Future research could explore first employing KV cache eviction to compress the cache to a certain proportion and then applying sparse attention for further acceleration. Ada-KV presented in this paper leverages head-wise adaptive allocation to enhance cache eviction methods, and this approach can similarly be applied in future work to enhance dynamic sparse attention methods.

---

[4]For additional related works, see Appendix A.1

# 3 Methodology

We begin by providing a formal description of a multi-head self-attention layer (Section 3.1). Building on this, we theoretically revisit the foundational principles of existing Top-$k$ eviction methods by introducing an $L_1$ eviction loss metric (Section 3.2). Inspired by theoretical findings, we propose a simple yet effective strategy for adaptive budget allocation, which is proven to outperform traditional uniform budget allocation (Section 3.3). We further demonstrate its compatibility with existing Top-$k$ eviction methods (Section 3.4) and present an efficient implementation (Section 3.5).

## 3.1 Preliminaries

LLMs operate through an autoregressive generation process, where each step relies on the last token to predict the next one. Let $X \in \mathbb{R}^{n \times d}$ represent the embedding matrix containing all tokens in the sequence, and let $x \in \mathbb{R}^{1 \times d}$ be the embedding of the last token used as input at the current time step. Using the notation system from [25] with $h$ attention heads, we provide a formal description of one multi-head self-attention layer. For each head $i \in [1, h]$, the transformation matrices $W_i^Q$, $W_i^K$, $W_i^V \in \mathbb{R}^{d \times d_h}$ map token embeddings to their respective Query, Key, and Value states, while the final output matrix $W_i^O \in \mathbb{R}^{d_h \times d}$ transforms the intermediate result to the output hidden states. At each time step, the previous KV cache elements of head $i$ are initialized as: $K_i = XW_i^K, V_i = XW_i^V$. Then, the embedding of input token $x$ is mapped to its respective Query, Key, and Value states for each head, and the previous KV cache is updated accordingly:

$$q_i = xW_i^Q, k_i = xW_i^K, v_i = xW_i^V \text{ and } K_i = Cat[K_i : k_i], V_i = Cat[V_i : v_i] \tag{1}$$

Finally, the output $y \in \mathbb{R}^{1 \times d}$ is computed using the attention weights $A_i \in \mathbb{R}^{1 \times n}$ as follows[5]:

$$y = \sum_{i \in [1,h]} A_i V_i W_i^O \text{ where } A_i = \text{softmax}(q_i K_i^T) \tag{2}$$

## 3.2 Theoretical Foundation: Revisiting Top-$k$ Methods with Bounded Eviction Loss

Top-k eviction methods typically presuppose the stability of critical cache elements during future generation process, to facilitate cache eviction [8, 11, 9, 10]. SOTA methods [11, 9, 10] commonly leverage query states of a series of tokens within an observation window to calculate observed attention weights with past KV cache elements. These weights, combined with Top-k selections, approximate the identification of cache elements critical in subsequent generations.

For ease of presentation, we assume a window size of 1, implying the eviction procedure relies solely on the attention weights $A$ between the past cache elements and the query state of the last token for the detection of critical cache elements. A set of indicator variables $\{\mathcal{I}_i \in \mathbb{R}^{1 \times n}\}^6$ represent the eviction decision, with allocated budgets $\{B_i\}$ for all heads $\{i \in [1, h]\}$, where $B = \sum_{i \in [1,h]} B_i$ is the overall budget for one attention layer:

$$\text{Eviction Decision:} \quad \mathcal{I}_i^j = \begin{cases} 1 & \text{Retain } K_i^j \text{ and } V_i^j, \\ 0 & \text{Evict } K_i^j \text{ and } V_i^j, \end{cases} \tag{3}$$

where $\mathcal{I}_i^j$ indicates whether the $j$th $\in [1, n]$ KV cache element in $K_i, V_i \in \mathbb{R}^{n \times d_h}$ is evicted for head $i$. Thus, only a budget size $B_i$ of cache elements is retained for head $i$: $\sum_{j \in [1,n]} \mathcal{I}_i^j = B_i$. Then, the post-eviction output $\hat{y}$ of multi-head self-attention mechanism is:

$$\hat{y} = \sum_{i \in [1,h]} \hat{A}_i V_i W_i^O, \text{where } \hat{A}_i = \text{softmax}(-\infty \odot (\mathbf{1} - \mathcal{I}_i) + q_i K_i^T), \tag{4}$$

and $\odot$ denotes element-wise multiplication.

The degradation in generation quality after cache eviction stems from changes in the attention output. We quantify the eviction loss as the $L_1$ distance between the pre- and post-eviction outputs of the self-attention mechanisms:

$$L_1 \text{ Eviction Loss} = ||y - \hat{y}||_1 \tag{5}$$

---

[5]The scaling factor $\sqrt{d_h}$ is omitted for simplification.

[6]Given that the first dimension of $\mathcal{I}_i$ is 1, $\mathcal{I}_i^j$ is used to simplify the notation for $\mathcal{I}_i(1, j)$. Similarly, $A_i^j$ follows the same notation.

Utilizing the row norm of the matrix, we derive an upper bound $\epsilon$ for the $L_1$ Eviction Loss in Theorem 3.1. For a detailed proof, please refer to Appendix A.8.

**Theorem 3.1.** *The $L_1$ eviction loss can be bounded by $\epsilon$:*

$$L_1 \text{ Eviction Loss} \leq \epsilon = 2hC - 2C \sum_{i \in [1,h]} \sum_{j \in [1,n]} \mathcal{I}_i^j A_i^j \tag{6}$$

*where $C = Max\left\{\|V_i W_i^O\|_\infty\right\}$ is a constant number, representing the max row norm .*

Essentially, existing cache eviction methods are equivalent to minimizing the upper bound $\epsilon$, as defined in Theorem 3.1. One such method, Top-$k$ cache eviction method [8, 11, 9, 10] stands out as a prominent approach, designed to selectively retain the most critical cache elements by prioritizing those with the highest attention weights. Specifically, the Top-$k$ eviction decision is formulated as :

$$\text{Top-}k \text{ Eviction Decision } \mathcal{I}_i^{*j} = \begin{cases} 1 \text{ if } A_i^j \in \text{Top-}k(A_i, k = B_i), \\ 0 \text{ Evict } K_i^j \text{ and } V_i^j. \end{cases}$$

This approach ensures that each head $i$ retains only the top $B_i$ critical cache elements based on attention weights $A_i^j$, evicting the rest. In Theorem 3.2, we prove that the Top-$k$ eviction decision $\{\mathcal{I}_i^*\}$ achieves the tightest upper bound $\epsilon^*$, establishing its effectiveness, under given budget allocation $\{B_i\}$. For a detailed proof, please refer to Appendix A.9.

**Theorem 3.2.** *The Top-$k$ cache eviction $\{\mathcal{I}_i^*\}$ minimizes the upper bound $\epsilon$ of $L_1$ eviction loss, resulting in $\epsilon^*$:*

$$\text{Top-}k \text{ eviction decision } \{\mathcal{I}_i^*\} = \underset{\{\mathcal{I}_i\}}{\arg\min}\, \epsilon \ \text{ and } \ \epsilon^* = \underset{\{\mathcal{I}_i\}}{\min}\, \epsilon = 2hC - 2C \sum_{i \in [1,h]} \sum_{j \in [1,n]}^{A_i^j \in Top\text{-}k(A_i, k=B_i)} A_i^j \tag{7}$$

Theorems 3.1 and 3.2 establish the theoretical basis for Top-$k$ eviction methods under any given budget allocation $\{B_i\}$. In the sequel, we show how an adaptive strategy can further minimize $\epsilon^*$, outperforming uniform budget allocation (i.e., $\{B_i = B/h\}$).

## 3.3 Optimizing Top-$k$ Methods with Adaptive Budget Allocation

---

**Algorithm 1** Ada-KV: Adaptive Budget Allocation

---

**Input**: Total budget: $B$; Attention weights for head $i$: $\{A_i\}$
**Output**: Allocated budgets $\{B_i^*\}$
1: Concatenate attention weights across heads $A = \text{Cat}(\{A_i\})$
2: Select top $B$ weights from $A$: Top-$k(A, k = B)$
3: Count number of selected weights for each head $i$: $\{f_i\}$
4: Set the allocated budgets as $\{B_i^* = f_i\}$
   **Return** allocated budgets $\{B_i^*\}$

---

Here, we propose the first adaptive budget allocation strategy for Top-$k$ eviction methods in Algorithm 1, designed to further minimize the upper bound $\epsilon^*$ in Theorem 3.2. It begins by identifying the $B$ largest attention weights across all heads within a single layer. Based on the frequency of selection in each head, the strategy dynamically determines the budget allocation $\{B_i^*\}$. Thereby, attention-sparse heads, where most attention weights are concentrated on only a few cache elements, are assigned a smaller budget. The saved budget is reallocated to attention-dispersed heads with more widespread concentration patterns. Thus it effectively adapts to the distinct attention patterns inherent to each head. Below, we further demonstrate the superiority of this strategy theoretically and empirically in subsequent sections.

**Theoretical Advantages of Adaptive Allocation.** Given the adaptive budget allocation $\{B_i^*\}$, the upper bound associated with the Top-$k$ eviction decision $\{\mathcal{I}_i^*\}$ is expressed as $\epsilon^{**}$:

$$\epsilon^{**} = 2hC - 2C \sum_{i \in [1,h]} \sum_{j \in [1,n]}^{A_i^j \in \text{Top-}k(A_i, B_i^*)} A_i^j \tag{8}$$

**Algorithm 2** Ada-SnapKV/Ada-Pyramid in One Layer

---

**Input**: total budget $B$, tokens in observation window $X^{win} \in \mathbb{R}^{win*d}$, cache in observation window $\{K_i^{win}, V_i^{win}\}$, cache outside observation window $\{K_i, V_i\}$

**Output**: retained cache $\hat{K}_i, \hat{V}_i$

1: **for** $i \leftarrow 1$ to $h$ **do**
2: $\quad Q_i^{win} = X^{win} W_i^Q$
3: $\quad \bar{A}_i = softmax(Q_i^{win} K_i^T)$
4: $\quad \bar{A}_i = \bar{A}_i.maxpooling(dim = 1).mean(dim = 0)$
5: **end for**
6: $B = B - winsize \times h$
7: Derive budget allocation $\{B_i^*\}$ using Algorithm 1$(B, \{\bar{A}_i\})$
8: Safeguard $\{B_i^*\} = \alpha \times \{B_i^*\} + (1 - \alpha) \times (B/h)$
9: Determine the Top-$k$ eviction decision $\{\mathcal{I}_i^*\}$ based on $\{B_i^*\}$
10: Select $\left\{\hat{K}_i, \hat{V}_i\right\}$ from $\{K_i, V_i\}$ according to $\{\mathcal{I}_i^*\}$
11: $\left\{\hat{K}_i, \hat{V}_i\right\} = \text{Cat}(\left\{\hat{K}_i, \hat{V}_i\right\}, \{K_i^{win}, V_i^{win}\})$
$\quad$ **Return** retained cache $\hat{K}_i, \hat{V}_i$

---

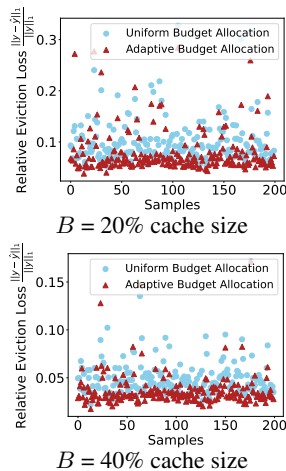

$B = 20\%$ cache size

$B = 40\%$ cache size

Figure 2: Adaptive allocation yields lower practical eviction losses (Llama-3.1-8B-Instruct on Qasper Dataset).

Theorem 3.3 demonstrates that the adaptive budget allocation achieves the minimum upper bound $\epsilon^{**}$ compared to any other allocation strategy, including the commonly used uniform allocation. A detailed proof is provided in Appendix A.10.

**Theorem 3.3.** *The adaptive budget allocation $\{B_i^*\}$, derived from Algorithm 1 achieves the minimal upper bound $\epsilon^{**}$ for loss associated with Top-$k$ eviction strategies:* $\epsilon^{**} = \min_{\{B_i\}} \epsilon^*$

Although a lower upper bound does not strictly guarantee reduced eviction loss, it is generally a key objective in algorithm design, as optimizing a tighter bound typically leads to better results.

**Empirical Evidence for Adaptive Allocation.** Empirically, the adaptive budget allocation described in Algorithm 1 assigns larger budgets to dispersed heads and conservatively adjust budgets for sparse heads. This approach aligns well with the significant variation in attention concentration across heads, as shown in Figure 1a. By adjusting the head-wise budget, this strategy effectively reduces practical eviction loss. Figure 2 provides a detailed visualization, showing that adaptive budget allocation consistently reduces practical eviction loss across most samples under the same cache budget. These results demonstrate the effectiveness of the proposed method in real-world scenarios.

### 3.4 Integration into Existing Cache Eviction Methods

We demonstrate the Plug-and-Play compatibility of Ada-KV strategy with two SOTA methods, SnapKV and Pyramid, by integrating it into their existing Top-$k$ eviction frameworks to create enhanced versions: Ada-SnapKV and Ada-Pyramid. Both SnapKV and Pyramid utilize tokens $X^{win} \in \mathbb{R}^{winsize*d}$ from a recent observation window (typically size 32) to identify and evict the less crucial elements in past KV cache. SnapKV excels under larger budget scenarios, while Pyramid is optimized for constrained budget conditions. This is because Pyramid uses a pyramidal budget distribution across attention layers via pre-set hyperparameters, prioritizing shallower layers. Thus, for a given layer with a total budget of $B$, their eviction algorithms are identical. However, both methods allocate budgets evenly across all heads within one layer.

Incorporating our adaptive allocation, as outlined in Algorithm 2, we modify these methods to better manage budget allocation at the head level. This integration occurs prior to the eviction process in each layer, where our strategy adaptively adjusts budget allocations based on the observed attention weights among heads, as shown in Line 7. Overall, they first calculate the observed attention weights $\bar{A}_i$ of past cache elements using the query states within the observation window. A max pooling layer processes these weights to preserve essential information [11], followed by a Top-$k$ selection of past cache elements outside the observation window. These selected elements, along with others within the observation window, are retained, while the rest are evicted. Moreover, we introduce a safeguard hyper-parameter, $\alpha$ (defaulted to 0.2), to prevent the allocation of excessively small budgets to highly sparse heads, thereby enhancing fault tolerance for presupposed critical stability [11, 8].

### 3.5 Implementation of Computation under Adaptive Budget Allocation

**Variable-length Attention with Variable-sized Cache Elements.** Adaptive allocation improves cache eviction quality but introduces challenges for efficient computation due to variable-sized cache elements across attention heads. We found that the variable-length FlashAttention technique [26, 27], widely adopted in many inference frameworks for continuous batching [28], could support efficient computation under adaptive allocation. To further enable this technique, we implement a flattened cache storage layout that concatenates the caches of all attention heads within a layer into a single tensor structure. This layout, combined with a custom CUDA kernel, enables efficient cache update operations. As demonstrated in Section 4.4, these components work in synergy to maintain computational efficiency under adaptive allocation, comparable to that of conventional FlashAttention.

**Compatibility with Group Query Attention.** Existing SOTA LLMs like Llama [29] and Mistral [30], have employed Group Query Attention (GQA) [31] technique to reduce KV cache sizes. However, existing cache eviction methods, such as SnapKV and Pyramid, lack GQA compatibility, redundantly replicating grouped KV caches across heads. We implement a simple GQA-compatible mechanism that uses the mean attention weight within each group as the selection criterion, eliminating redundancy. This enables SnapKV, Pyramid, and their adaptive variants—Ada-SnapKV and Ada-Pyramid—to achieve significant cache size reductions, such as a 4x reduction in Llama-3.1-8B.

## 4 Experiments

### 4.1 Settings

**Base Models.** We employ two open-source base models: Llama-3.1-8B-Instruct [29] and Mistral-7B-instruct-v0.2 [30]. These models, widely adopted due to moderate parameter sizes and impressive performance on long-sequence tasks, both leverage the GQA [31] technique, which reduces the KV cache size to one-quarter of its original.

**Datasets.** We conduct evaluations using two popular benchmarks: Ruler [32] and LongBench [33]. Ruler includes 13 long-sequence tasks, primarily variations of the Needle-in-a-Haystack test, offering a range of difficulty levels for comprehensive model assessment. According to the official evaluation [32], the maximum effective length for Mistral-7B-instruct-v0.2 and Llama-3.1-8B-Instruct with the full KV Cache is 16K. Therefore, we select the 16K Ruler Benchmark as the primary benchmark for evaluation. Additionally, we also conduct evaluations using LongBench, including 16 datasets covering multiple task domains. Detailed dataset information is provided in Appendix A.5 and A.7.

**Baselines.** We select the *SnapKV*[11] and *Pyramid*[9, 10] as the primary baselines, given that they are SOTA methods and foundational bases for our *Ada-SnapKV* and *Ada-Pyramid* methods. All these methods implement GQA compatibility as stated in Section 3.5, reducing cache size to a quarter of previous naive implementations. Additionally, *StreamingLLM* [16] is also included as a representative of Sliding Window Eviction Methods for reference.

**Parameters.** Cache eviction methods can be applied whenever KV cache compression is required. Following standard practices in prior studies [11, 9, 10], we perform cache eviction after the prefilling phase of each layer for consistent comparison. In all experiments, the hyperparameter $\alpha$ for adaptive budget allocation is fixed at 0.2. Both *Ada-SnapKV* and *Ada-Pyramid*, as well as *SnapKV* and *Pyramid*, follow the configuration settings outlined in [11], including an observation window size of 32 and a max pooling kernel size of 7.

### 4.2 Ruler Benchmark

We evaluate quality scores for each method using budgets set to 20%, 40%, 60%, and 80% of the original cache size. Beyond the commonly studied **question-aware** compression scenario—where questions are known in advance—we also consider the more challenging **question-agnostic** setting [12], in which questions are revealed only after compression. This mimics more challenging scenarios, such as prompt caching [34, 35], where numerous question-independent prefix prompts need compression, or multi-turn dialogue [1], where compression occurs without future questions.

**Overall Results.** Figure 3 presents the average scores across 13 datasets in the Ruler Benchmark. Overall, existing Top-k eviction methods exhibit significant performance drops in the question-

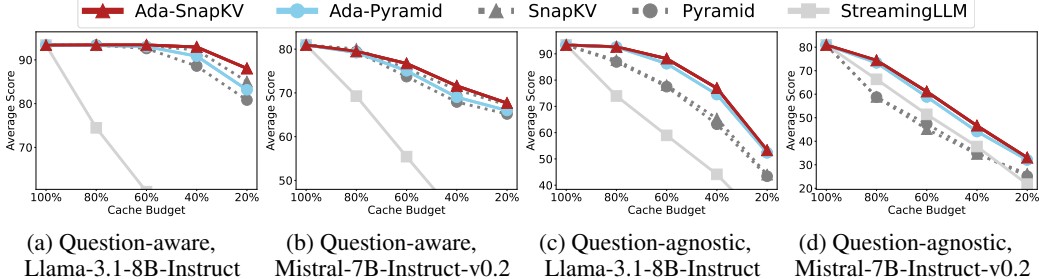

(a) Question-aware, Llama-3.1-8B-Instruct  (b) Question-aware, Mistral-7B-Instruct-v0.2  (c) Question-agnostic, Llama-3.1-8B-Instruct  (d) Question-agnostic, Mistral-7B-Instruct-v0.2

Figure 3: Average Score on Ruler Among 13 Datasets.

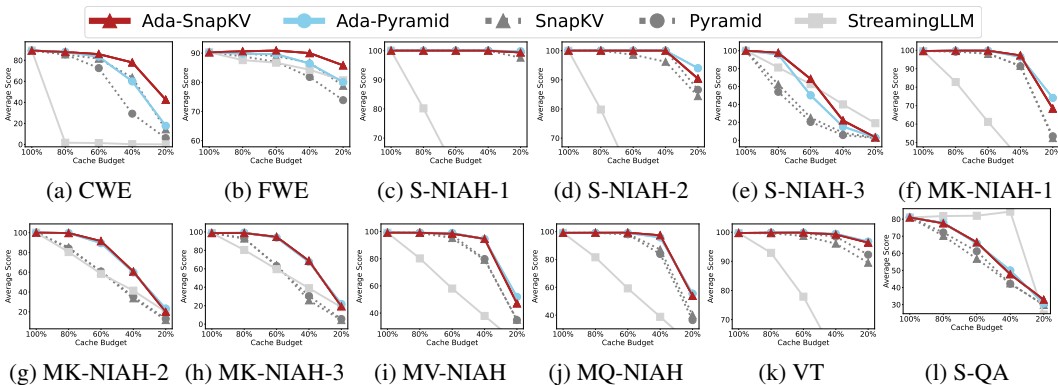

(a) CWE  (b) FWE  (c) S-NIAH-1  (d) S-NIAH-2  (e) S-NIAH-3  (f) MK-NIAH-1

(g) MK-NIAH-2  (h) MK-NIAH-3  (i) MV-NIAH  (j) MQ-NIAH  (k) VT  (l) S-QA

Figure 4: Subtask Analysis on Ruler (Question-agnostic, Llama-3.1-8B-Instruct).

Table 1: Task Analysis for Llama-3.1-8B (Question-agnostic).

| Task Domains | Full Cache | $b = 10\%$ | | $b = 20\%$ | | $b = 40\%$ | |
| --- | --- | --- | --- | --- | --- | --- | --- |
| | | SnapKV | Ada-SnapKV | SnapKV | Ada-SnapKV | SnapKV | Ada-SnapKV |
| Single-Doc. QA | 43.10 | 23.38 | **23.97** | 28.78 | **31.39** | 35.27 | **36.63** |
| Multi-Doc. QA | 46.49 | 26.61 | **29.06** | 33.51 | **34.90** | 40.50 | **41.36** |
| Summarization | 28.97 | 21.98 | **22.43** | 23.82 | **24.29** | 26.11 | **26.66** |
| Few-shot | 69.45 | 57.89 | **60.40** | 61.95 | **63.70** | 65.10 | **66.43** |
| Synthetic | 53.73 | **36.50** | 34.88 | 48.19 | **50.39** | **53.17** | 53.00 |
| Code | 57.86 | 59.72 | **60.91** | 60.05 | **61.14** | **60.49** | 60.30 |
| Ave. | 49.20 | 36.38 | **37.45** | 41.29 | **42.87** | 45.52 | **46.24** |

agnostic scenario. In contrast, our Ada-SnapKV and Ada-Pyramid consistently outperform the original SnapKV and Pyramid across both question-aware and question-agnostic scenarios on the two LLMs. Using the Llama-3.1-8B-Instruct model as an example: in the simpler question-aware scenario (Figure 3a), both Ada-SnapKV and Ada-Pyramid significantly reduce quality loss under small compression budgets, such as 40% and 20% cache sizes, compared to the original SnapKV and Pyramid. In the more challenging question-agnostic scenario (Figure 3c), Ada-SnapKV and Ada-Pyramid substantially reduce quality loss across all cache budget settings. For instance, leveraging the adaptive allocation strategy, Ada-SnapKV improves SnapKV's scores at 80% and 20% cache sizes from 87.59 and 44.02 to 92.67 and 53.29, respectively.

**Subtask Analysis.** Figure 4 presents the results for 12 datasets using Llama-3.1-8B-Instruct in the challenging question-agnostic scenario, highlighting the improvements brought by the adaptive budget allocation strategy. More results are provided in Appendix A.2, where our method also shows significant improvements. Overall, our adaptive budget allocation strategy significantly enhances the performance of existing methods. For example, at 80% cache budget, the original SnapKV demonstrates noticeable degradation on many tasks, whereas Ada-SnapKV maintains near-lossless performance across most tasks. Particularly in difficult Needle-in-a-Haystack tasks like S-NIAH-3 and MK-NIAN-2 with 80% cache budget, Ada-SnapKV increases SnapKV's scores from 62.4 and 85.2 to 97.6 and 99.6, as shown in Figures 4e and 4g. These results emphasize the effectiveness of incorporating adaptive budget allocation for enhancement, especially in difficult tasks.

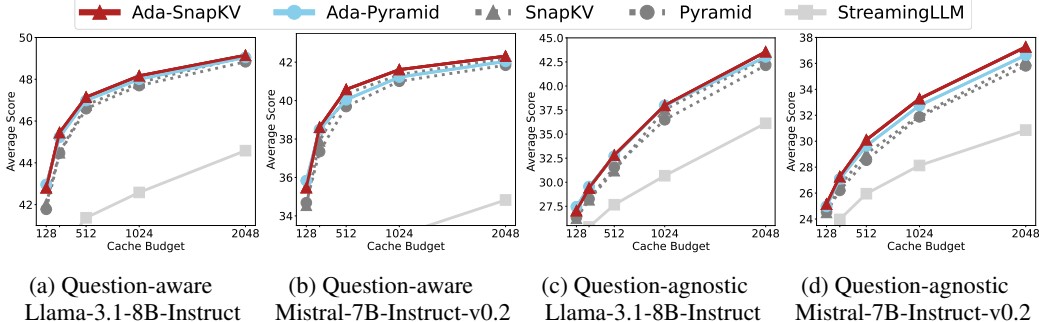

Figure 5: Average Score on LongBench among 16 Datasets under Fixed Budgets.

## 4.3 LongBench Benchmark

To align with previous evaluations [11, 10], we first set the average budget for each head to fixed values of $\{128, 256, 512, 1024, 2048\}$. We further extend from standard question-aware compression to question-agnostic scenarios by separating the questions in Longbench, thereby better reflecting real-world performance, as suggested by [12]. For more details, please refer to Appendix A.7.

**Question-aware Compression:** Figures 5a and 5b present the average scores across 16 datasets under fixed budget constraints with the question-aware compression. [7]. SnapKV and Pyramid, both utilizing Top-$k$ selection within an observation window, achieve closely matched performance and outperform the previous sliding window eviction method, StreamingLLM. Moreover, our Ada-SnapKV and Ada-Pyramid methods consistently improve generation quality across different budgets, alternately outperforming their respective base methods and, under a fixed budget of 2048, approaching lossless performance. These consistent gains highlight the effectiveness of adaptive budget allocation.

**From Question-aware to Question-agnostic Compression:** Figures 5c and 5c further present the average scores with the question-agnostic compression. Our enhanced Ada-SnapKV and Ada-Pyramid methods continue to outperform the original SnapKV and Pyramid methods. However, a comparison between question-aware and question-agnostic settings reveals a notable performance drop across all methods. For example, with Llama and a 2048 budget, the average score of SnapKV decreases from 49.09 to 42.86 while transfer from question-aware to question-agnostic settings. This indicates that commonly used question-aware settings do not adequately capture the limitations of cache eviction methods in a more realistic compression scenario. We therefore suggest that future evaluations of cache eviction techniques should place greater emphasis on question-agnostic compression to improve robustness in practical applications.

**More Evaluation in Question-agnostic Compression:** To more accurately assess question-agnostic compression, we apply ratio-based cache budget compression, which better reflects real-world performance on Longbench's highly imbalanced sample lengths. [8] Table 1 provides a detailed breakdown of performance across task domains for Llama-3.1-8B. Overall, Code tasks are largely insensitive to cache compression, and in some cases, even see improved performance after compression. In contrast, other tasks experience significant degradation, particularly in Summarization and QA. Notably, our Ada-SnapKV effectively mitigates these losses and improves overall performance. Across 18 domain cases and three cache budgets, Ada-SnapKV delivers quality improvements in 15 domains, demonstrating its robust benefits. Table 2 further reports results on the larger Llama-3.1-70B model, showing similarly strong gains from our approach. Except for the Code domain, which remains insensitive to compression, significant improvements are observed in all other tasks.

## 4.4 Computation Efficiency Under Adaptive Allocation

Cache eviction methods aim to enhance the computational (memory and runtime) efficiency by selectively evicting KV cache elements. We demonstrate that, supported by our flattened cache layout and custom CUDA kernels, the computational efficiency of cache eviction under adaptive

---

[7]Detailed scores can be found in Appendix A.4

[8]As shown in Table 12 (Appendix A.7), Longbench sample lengths are highly imbalanced—for example, the Code task Lcc averages 1,235 tokens, while the QA task NarrativeQA averages 18,409. Thus, applying a fixed budget, like 2048, across all datasets may introduce bias when evaluating cache eviction across tasks.

Table 2: Task Analysis for Llama-3.1-70B (Question-agnostic).

| Domain | Full Cache | $b = 20\%$ | | $b = 40\%$ | |
|---|---|---|---|---|---|
| | | SnapKV | Ada-SnapKV | SnapKV | Ada-SnapKV |
| Single-Doc. QA | 47.15 | 32.54 | **36.05** | 39.35 | **42.21** |
| Multi-Doc. QA | 60.07 | 46.98 | **48.45** | 54.07 | **55.03** |
| Summarization | 28.69 | 24.40 | **24.81** | 26.21 | **26.57** |
| Few-shot | 72.07 | 66.71 | **67.80** | 69.29 | **70.29** |
| Synthetic | 58.25 | 51.25 | **51.50** | 56.00 | **56.75** |
| Code | 47.82 | **52.44** | 51.51 | **51.90** | 49.23 |
| Ave. | 52.25 | 44.95 | **46.08** | 48.91 | **49.64** |

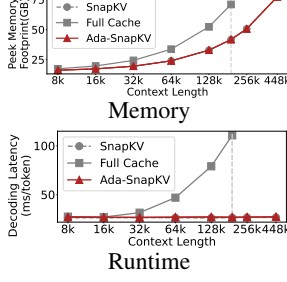

Figure 6: Efficiency (All with FlashAttention-2)

Table 3: Performance gains from applying the Ada-KV strategy to follow-up methods. All results are from the Llama-3.1-8B model on the Longbench benchmark under question-agnostic settings.

| Cache | DuoAttn *Training-based* | HeadKV *Training-based* | CriticalKV w/o. Ada-KV | CriticalKV w/. Ada-KV | DefensiveKV w/o. Ada-KV | DefensiveKV w/. Ada-KV |
|---|---|---|---|---|---|---|
| 20% | 39.52 | 42.64 | 42.99 | 43.77 | 43.78 | **46.68** |
| 40% | 48.17 | 47.23 | 47.29 | 48.00 | 47.76 | **49.21** |

allocation, based on the variable-length FlashAttention technique, matches that of traditional uniform eviction methods within the same cache budget constraints. As shown in Figure 6, with a fixed budget of 1024, Ada-SnapKV achieves peak memory usage and decoding latency comparable to the original SnapKV, and both significantly outperform the full cache case. These results demonstrate that Ada-KV strategy effectively enhance post-eviction generation quality while maintaining strong computational efficiency.

## 5 Broad Benefits of the Adaptive Budget Allocation Strategy

Due to its plug-and-play design, Ada-KV is broadly applicable beyond the two cases presented in this paper. As of publication, many follow-up works have adopted Ada-KV strategy, yielding a wide range of enhanced applications [36, 37, 38], especially CriticalKV [39] and DefensiveKV [40]. Beyond direct integration, concurrent methods have explored budget allocation through training-based profiling. For instance, DuoAttention [41] categorizes heads into "full attention" and "streaming" types, while HeadKV [42] enables finer-grained allocation building on our open-source implementation. We present the results of these follow-up studies in Table 3 to demonstrate the broad benefits of adaptive budget allocation strategy. When combined with CriticalKV and DefensiveKV, Ada-KV consistently improves performance, which in turn scales with the strength of the base method. Notably, even the plug-and-play CriticalKV and DefensiveKV methods, when enhanced with the Ada-KV strategy, surpass the performance of training-based approaches like DuoAttention and HeadKV. These findings demonstrate that Ada-KV retains substantial value and offers significant enhancement potential, even alongside more recent and advanced methods.

## 6 Conclusion

In this study, we revisit cache eviction strategies for efficient LLM inference and uncover a key overlooked factor: adaptive budget allocation across attention heads. Guided by a theoretical analysis of the loss upper bound, we propose *Ada-KV*, the first adaptive budget allocation strategy for optimizing KV cache eviction methods. To demonstrate its plug-and-play benefits, we seamlessly integrate Ada-KV into two existing SOTA methods, introducing Ada-SnapKV and Ada-Pyramid, two adaptive eviction methods. Beyond the commonly studied question-aware compression, we also evaluate these methods in the more challenging and less explored question-agnostic compression scenario. Using the Ruler and LongBench benchmarks, we demonstrate the effectiveness of adaptive budget allocation in both settings. Our results not only expose the limitations of current cache eviction strategies but also highlight the potential of adaptive allocation to enhance cache eviction.

## Acknowledgments

We would like to thank the anonymous reviewers for their valuable feedback and constructive comments. This work was supported by the National Natural Science Foundation of China (NSFC) under Grants 62472400 and 62271465, the National Key R&D Program of China under Grant 2025YFC3408300, and the Suzhou Basic Research Program under Grant SYG202338.

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

# A   Appendix

## A.1   Additional Related Works

Additional works also mitigate the challenges posed by massive KV Caches during long-sequence inference while not reducing the number of cache elements. These works are orthogonal to our work. For instance, in our implementation, we have integrated the Flash Attention [26] technique to enhance efficient computation. Similar efforts, such as Paged Attention [28], employ efficient memory management strategies to reduce I/O latency without altering the size of the KV Cache. KV cache quantization methods [43, 44, 45], reduce the size of cache by lowering the precision of individual elements. The cache eviction techniques focused in this paper also be further combined and complemented with quantization in the future. Some studies also employ speculative decoding to accelerate long-sequence generation [46, 47, 48], typically using model with a reduced KV cache to generate drafts. Future work could integrate more advanced cache eviction methods to further improve the efficiency of such approaches.

## A.2   Detail results of Ruler Evaluation

Figure 7 complements the missing result for the multi-hop QA subtask in Figure 4 due to the space limitation. Figures 9 8 10 further provide a comprehensive overview of the performance of Ada-SnapKV and Ada-Pyramid across all 16K Ruler subtasks on two LLMs in both question-agnostic and question-aware scenarios. The results demonstrate that regardless of the scenario (question-agnostic or question-aware), model (LLama or Mistral), or cache size budgets, Ada-SnapKV and Ada-Pyramid outperform the original SnapKV and Pyramid approaches in most tasks. This highlights the broad applicability and effectiveness of the adaptive allocation strategy.

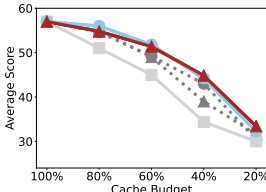

Figure 7: M-QA Subtask Analysis on Ruler for Question-agnostic Scenario (Llama3.1-8B-Instruct)

## A.3   Robustness Analysis of Safeguard $\alpha$

Rather than fine-tuning the safeguard parameter, $\alpha$, for each model or budget—a process that would introduce considerable complexity—we opted for a fixed value. As shown in Table [Table Number], we conducted a robustness analysis of $\alpha$ using Mistral-7B on the LongBench benchmark. The results indicate that a smaller $\alpha$ allows for more aggressive budget allocation, improving performance under limited budgets. Conversely, a larger $\alpha$ performs slightly better with higher budgets. Based on this trade-off, we selected a balanced value for $\alpha = 0.2$ in this work.

Table 4: Robustness Analysis of $\alpha$ (Question-aware)

| Longbench, Mistral-7B | Budget 128 | Budget 256 | Budget 512 | Budget 1024 |
|---|---|---|---|---|
| Ada-SnapKV $\alpha = 0$ | **35.69** | **38.9** | 40.53 | 41.44 |
| Ada-SnapKV $\alpha = 0.2$ | 35.48 | 38.61 | 40.58 | 41.61 |
| Ada-SnapKV $\alpha = 0.5$ | 35.2 | 38.46 | **40.66** | **41.62** |

## A.4   Detail results of LongBench Evaluation

Tables 5, 6, 8 and 7 present the detailed evaluation results of different methods across 16 datasets on LongBench for two models. As shown, the Ada-SnapKV and Ada-Pyramid methods, with our adaptive allocation enhancement, achieve consistent improvements in overall generation quality across different budgets and models.

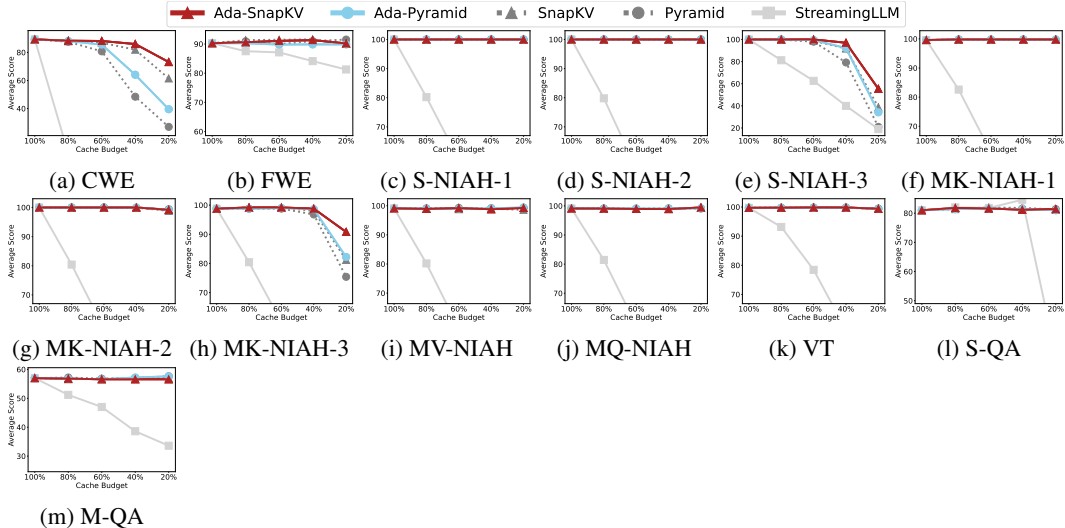

Figure 8: Subtask Analysis on Ruler (Question-aware, Llama3.1-8B-Instruct).

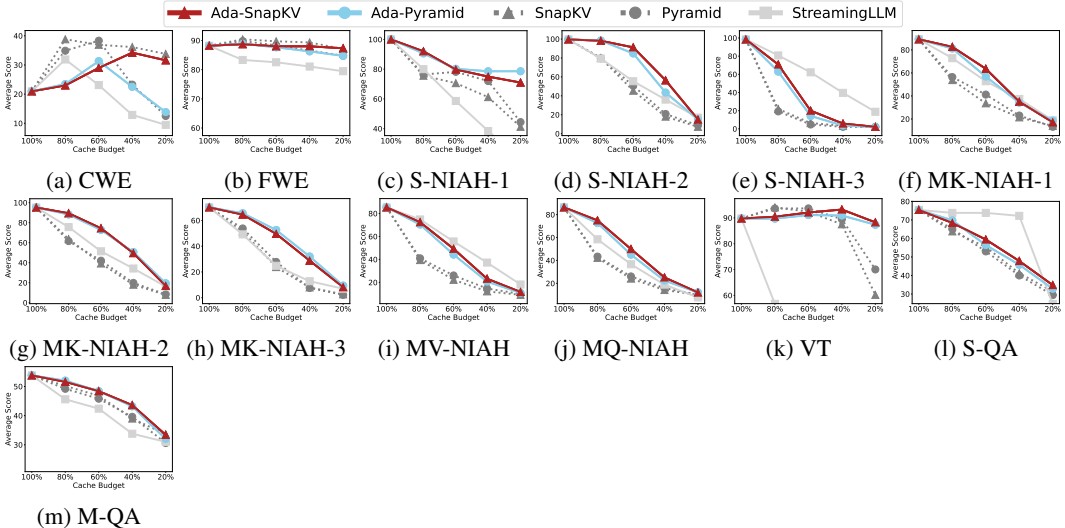

Figure 9: Subtask Analysis on Ruler (Question-agnostic, Mistral-7B-Instruct-v0.2).

## A.5    Detailed Information of Ruler Benchmark

Below is a brief overview of the different tasks in Ruler. For detailed examples, please refer to Tables 9 10 and 11. We adhere to the input prompt format from KVPress [12], dividing the input into context and question segments. The question segment is highlighted in green, while other colors represent the context segment. In question-aware compression, both the context and question segments are input into the model and compressed. For question-agnostic compression, where future questions are unpredictable, only the context segment is input for compression. After compression, the question segment is input for answer generation.

- **Single NIAH (S-NIAH):** A basic information retrieval task. In this scenario, a keyword sentence, referred to as the "needle," is embedded within a lengthy text, called the "haystack." The goal is to retrieve the "needle" from the context. The "haystack" may consist of repetitive, noisy sentences or essays by Paul Graham [49].
- **Multi-keys NIAH (MK-NIAH):** Multiple "needles" are inserted into the "haystack," but the task requires retrieving only one of them.

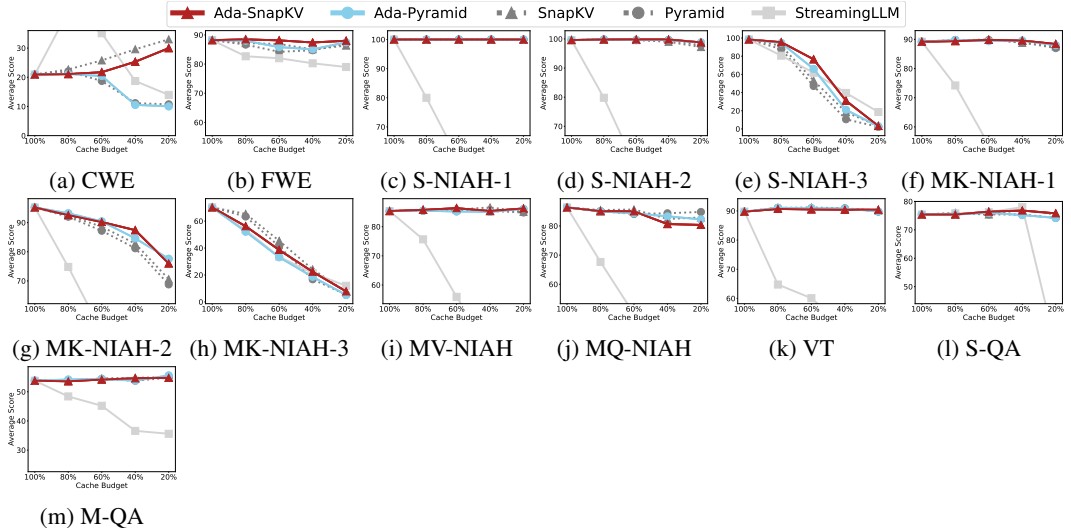

Figure 10: Subtask Analysis on Ruler (Question-aware, Mistral-7B-Instruct-v0.2).

- **Multi-values NIAH (MV-NIAH):** Multiple "needles" in the form of key-value pairs are placed within the "haystack." The task is to retrieve all values associated with a given key.
- **Multi-queries NIAH (MQ-NIAH):** Multiple "needle" in the form of key-value pairs are inserted into the "haystack". All "needles" corresponding to the keys specified in the queries need to be retrieved. **(author?)** [50]
- **Variable Tracking (VT):** A series of variable binding statements (e.g., $X2 = X1, X3 = X2, ...$) are inserted at various positions within a long text to simulate the recognition of entities and relationships. The task objective is to return all variable names that point to the same value $V$.
- **Common words extraction task (CWE):** The input sequence is constructed by sampling words from a discrete uniform distribution within a pre-defined (synthetic) word list. Words that appear most frequently are termed "common words". The model is required to identify all common words. The number of common words remains constant while the number of uncommon words increases with the length of the sequence.
- **Frequent words extraction task (FWE):** The input sequence is constructed by sampling words from a Zeta distribution within a pre-defined (synthetic) word list. The model need to return the top-K frequent words in the context.
- **Single-Hop QA (S-QA)** Answer questions based on passages. The context is extended to simulate long-context input by adding distracting information(golden paragraphs). And answers come from a single piece of evidence.
- **Multi-Hop QA (M-QA)** Answer questions based on passages. For Multi-hop QA, which requires integrating information from multiple sources.

### A.6 Limitations

This paper, for the first time, introduces a head-wise allocation strategy for KV cache compression, supported by both theoretical analysis and empirical validation. However, this head-wise allocation is still limited to within single layer. In future work, we plan to extend the head-wise allocation mechanism across layers and further develop corresponding theoretical analysis to demonstrate advantages.

### A.7 Detailed Information of LongBench Benchmark

LongBench [33] spans multiple task domains with an average length of 6711, including single-document QA [51, 52], multi-document QA [53, 54, 55], summarization [56, 57, 58], few-shot learning [59, 60, 61], synthetic tasks [33], and code generation [62, 63].

- **Single-Doc QA:** Single-document QA requires models to obtain the answer from a single piece of source. The test samples are derived from diverse datasets including NarrativeQA [51], qasper [52], and MultiFieldQA [64], encompassing a range of documents such as legal files, governmental reports, encyclopedia, and academic papers.
- **Multi-Doc QA:** Multi-document QA requires models to extract and combine information from several documents to obtain the answer. The test samples are built from three Wikipedia-based multi-hop QA datasets: HotpotQA [53], 2WikiMultihopQA [65] and MuSiQue [55].
- **Summarization:** Summarization task requires a comprehensive understanding of the context. The test samples are drawn from the GovReport dataset [56], and QMSum [57]. Additionally, the MultiNews dataset originates from a multi-document summarization corpus detailed in [58]
- **Few-shot Learning:** Few-shot Learning task have integrated classification, summarization, and reading comprehension tasks to maintain task diversity. For classification, the TREC dataset [61] is included. The SAMSum dataset [60], which comprises messenger-style conversations with human-annotated summaries, has been incorporated for summarization tasks. Additionally, TriviaQA [66], a dataset of question-answer pairs, is utilized for reading comprehension tasks.
- **Synthetic Task:** Synthetic Tasks are designed to assess the model's ability in handling specific scenarios and patterns. Two synthetic tasks, PassageRetrieval-en and PassageCount, are included in LongBench. PassageRetrieval-en is derived from English Wikipedia. 30 passages is randomly selected and use GPT-3.5-Turbo to summarize one of them. The task challenges the model to identify the original paragraph that matches the generated summary. PassageCount is designed to be more challenging, this task requires the model to leverage the entire context to solve it. Several passages are randomly pickedfrom English Wikipedia, randomly duplicate each paragraph multiple times, and then shuffle the paragraphs. The task is to determine the number of unique passages within the provided set.
- **Code Completion:** Code Completion task is to assist users by completing code based on previous code input and context. The LCC dataset, derived from the original Long Code Completion dataset [67]. The RepoBench-P dataset [68] is designed to aggregating information from code across files. Model is required to predict the next line of code.

Table 12 provides a comprehensive description of information pertaining to 16 datasets in LongBench. In the question-agnostic scenarios, we separate the questions from the context, with the question portions highlighted in Tables 13, 14, 15, 16, 17, and 18.

## A.8 Proof of Theorem 3.1

**Theorem A.1.** *The $L_1$ eviction loss can be bounded by $\epsilon$:*

$$L_1 \ Eviction \ Loss \leq \epsilon = 2hC - 2C \sum_{i \in [1,h]} \sum_{j \in [1,n]} \mathcal{I}_i^j A_i^j \tag{9}$$

*where $C = Max \left\{ \|V_i W_i^O\|_\infty \right\}$ is a constant number, representing the max row norm among all matrices.*

*Proof.* Consider the softmax function as:

$$softmax(x)^j = \frac{exp(x^j)}{\sum_j exp(x^j)} \tag{10}$$

Thus, the attention weight after eviction procedure is:

$$\hat{A}_i = \text{softmax}(-\infty \odot (\mathbf{1} - \mathcal{I}_i) + s_i) \ \text{where} \ s_i = q_i K_i^T \tag{11}$$

$$\hat{A}_i^j = \frac{exp(s_i^j - \infty \odot (1 - \mathcal{I}_i^j))}{\sum_j exp(s_i^j - \infty \odot (1 - \mathcal{I}_i^j))} \tag{12}$$

$$= \frac{\mathcal{I}_i^j exp(s_i^j)}{\sum_j \mathcal{I}_i^j exp(s_i^j)} = \frac{\mathcal{I}_i^j exp(s_i^j)}{\sum_j exp(s_i^j)} \frac{\sum_j exp(s_i^j)}{\sum_j \mathcal{I}_i^j exp(s_i^j)} \tag{13}$$

Table 5: Detailed results of Llama-3.1-8B-Instruct on LongBench.(Question-aware)

| | Single-Doc. QA | | | Multi-Doc. QA | | | Summarization | | | Few-shotLearning | | | Synthetic | | Code | | Ave. Score |
|---|---|---|---|---|---|---|---|---|---|---|---|---|---|---|---|---|---|
| | NrtvQA | Qasper | MF-en | HotpotQA | 2WikiMQA | Musique | GovReport | QMSum | MultiNews | TREC | TriviaQA | SAMSum | PCount | PRe | Lcc | RB-P | |
| Full Cache | 30.22 | 45.37 | 55.80 | 55.97 | 45.00 | 31.26 | 35.12 | 25.38 | 27.20 | 72.50 | 91.64 | 43.57 | 9.41 | 99.50 | 62.88 | 56.43 | 49.20 |
| **B=128** | | | | | | | | | | | | | | | | | |
| SLM | 22.24 | 20.87 | 31.72 | 44.02 | 37.55 | 24.54 | 18.76 | 21.09 | 18.48 | 40.50 | 84.41 | 38.82 | 8.00 | 99.50 | 57.02 | 47.29 | 38.43 |
| Pyramid | 25.70 | 24.69 | 47.74 | 52.87 | 40.57 | 27.23 | 20.02 | 22.38 | 19.74 | 44.50 | 88.81 | 40.30 | 7.22 | **99.50** | 57.25 | 49.90 | 41.78 |
| SnapKV | 25.54 | 24.45 | 48.03 | **53.31** | 40.75 | 28.19 | 20.13 | 22.36 | 19.55 | 45.50 | 89.20 | 40.62 | 6.97 | **99.50** | 58.45 | 49.90 | 42.03 |
| Ada-Pyramid | **27.07** | **25.61** | 49.30 | 53.02 | 41.29 | 27.83 | **20.70** | 23.18 | **20.38** | 51.50 | **90.76** | 40.62 | 6.92 | 99.00 | **59.30** | 50.88 | **42.96** |
| Ada-SnapKV | 24.90 | 24.41 | **49.95** | 53.15 | **41.73** | **28.55** | 20.54 | **23.21** | 20.28 | 50.50 | 89.49 | **40.71** | 7.45 | 99.00 | 58.74 | **52.40** | 42.81 |
| **B=256** | | | | | | | | | | | | | | | | | |
| SLM | 22.71 | 23.79 | 31.80 | 43.43 | 36.55 | 25.55 | 21.29 | 20.68 | 20.67 | 46.00 | 87.11 | 40.82 | 7.20 | 99.50 | 59.89 | 49.19 | 39.76 |
| Pyramid | 25.53 | 33.15 | 51.44 | **55.03** | 42.42 | 28.62 | 22.57 | 23.37 | 22.33 | 56.50 | 91.19 | **41.28** | 6.97 | 99.50 | 60.36 | 51.18 | 44.47 |
| SnapKV | 26.02 | 32.49 | 51.62 | 54.40 | **42.77** | 28.94 | 22.83 | 23.54 | 22.55 | 53.50 | 91.10 | 40.95 | 7.48 | 99.50 | 60.67 | 53.39 | 44.48 |
| Ada-Pyramid | 25.12 | **35.06** | **52.28** | 54.66 | 41.89 | 28.76 | 23.14 | 23.36 | 22.67 | 63.00 | 90.72 | 41.21 | 7.75 | 99.50 | 61.47 | 53.09 | 45.23 |
| Ada-SnapKV | **26.11** | 33.39 | 51.44 | 54.94 | 42.15 | **29.54** | 23.01 | **23.85** | 22.88 | **63.50** | **91.57** | 40.94 | **8.00** | 99.50 | **61.95** | **54.33** | **45.44** |
| **B=512** | | | | | | | | | | | | | | | | | |
| SLM | 25.51 | 25.78 | 34.19 | 45.01 | 35.91 | 24.93 | 23.61 | 21.26 | 23.57 | 57.50 | 87.86 | 41.44 | 6.98 | 96.50 | 60.85 | 51.02 | 41.37 |
| Pyramid | 28.71 | 39.89 | 52.86 | 54.00 | **44.20** | 31.22 | 24.74 | 23.73 | 24.28 | 66.00 | 91.07 | 41.42 | **8.39** | 99.50 | 61.99 | 53.44 | 46.59 |
| SnapKV | **29.22** | 40.01 | **53.15** | 54.47 | 43.63 | **31.32** | 25.04 | 23.77 | 24.19 | 64.00 | 92.05 | 41.57 | 8.01 | 99.50 | 63.21 | 55.05 | 46.76 |
| Ada-Pyramid | 28.04 | **40.63** | 53.03 | **54.71** | 43.39 | 30.26 | 25.35 | 24.12 | 24.61 | **69.00** | 91.79 | **42.55** | 7.95 | 99.50 | 62.28 | 54.49 | 46.98 |
| Ada-SnapKV | 29.07 | 40.16 | 52.44 | 53.90 | 43.05 | 31.10 | **25.75** | **24.39** | **24.85** | 69.00 | **92.34** | 42.05 | 7.98 | 99.50 | **63.43** | **55.32** | **47.15** |
| **B=1024** | | | | | | | | | | | | | | | | | |
| SLM | 24.97 | 30.22 | 37.06 | 46.57 | 39.14 | 25.24 | 26.01 | 21.08 | 25.72 | 63.50 | 88.87 | 42.28 | 6.98 | 89.00 | 61.30 | 53.40 | 42.58 |
| Pyramid | **29.62** | 43.66 | 54.10 | **55.06** | 44.22 | 31.30 | 27.27 | 24.30 | 25.68 | 68.50 | 91.27 | 41.96 | 7.73 | **99.50** | 63.13 | 55.85 | 47.70 |
| SnapKV | 29.28 | 43.64 | **54.34** | 54.24 | **44.34** | 31.52 | 27.80 | 24.39 | 25.61 | 69.00 | **91.72** | 42.50 | 7.80 | 99.50 | 62.99 | 56.45 | 47.84 |
| Ada-Pyramid | 28.76 | **44.57** | 53.73 | 54.89 | 44.15 | **31.97** | 27.75 | **25.26** | 25.84 | 70.50 | 91.62 | 42.37 | 7.67 | 99.50 | 62.96 | **56.52** | 48.00 |
| Ada-SnapKV | 29.23 | 44.09 | 53.82 | 54.80 | 44.01 | 31.40 | **28.86** | 24.73 | **26.04** | **72.50** | 91.72 | **42.56** | 7.82 | 99.50 | 63.22 | 56.33 | **48.16** |
| **B=2048** | | | | | | | | | | | | | | | | | |
| SLM | 26.25 | 35.81 | 38.78 | 48.71 | 41.78 | 25.01 | 28.48 | 22.13 | 26.55 | 67.50 | 90.86 | 42.21 | 9.41 | 87.50 | 64.80 | 57.56 | 44.58 |
| Pyramid | 29.81 | 45.13 | 54.88 | 55.74 | 44.36 | **31.71** | 30.28 | 24.83 | 26.64 | 71.00 | 91.35 | 42.42 | **10.43** | 99.50 | 64.66 | 58.64 | 48.84 |
| SnapKV | 29.75 | 45.18 | 55.06 | **56.09** | 44.82 | 31.42 | 31.13 | 24.92 | 26.93 | 72.00 | **91.65** | 42.81 | 9.99 | 99.50 | 64.82 | **59.30** | 49.04 |
| Ada-Pyramid | **30.88** | 44.64 | 54.94 | 55.59 | 45.00 | 31.04 | 30.30 | **25.14** | 26.93 | 72.00 | 91.64 | **43.25** | 10.09 | 99.50 | 64.60 | 59.20 | 49.05 |
| Ada-SnapKV | 30.64 | **45.23** | **55.46** | 55.55 | **45.26** | 31.14 | **31.28** | 24.89 | 26.72 | **72.50** | 91.64 | 42.75 | 9.68 | **99.50** | **64.88** | 59.25 | **49.15** |

Table 6: Detailed results of Mistral-7B-Instruct-v0.2 on LongBench. (Question-aware)

| | Single-Doc. QA | | | Multi-Doc. QA | | | Summarization | | | Few-shotLearning | | | Synthetic | | Code | | Ave. Score |
|---|---|---|---|---|---|---|---|---|---|---|---|---|---|---|---|---|---|
| | NrtvQA | Qasper | MF-en | HotpotQA | 2WikiMQA | Musique | GovReport | QMSum | MultiNews | TREC | TriviaQA | SAMSum | PCount | PRe | Lcc | RB-P | |
| Full Cache | 26.74 | 32.84 | 50.00 | 43.45 | 27.77 | 18.49 | 32.91 | 24.64 | 26.99 | 71.00 | 86.23 | 43.32 | 2.94 | 86.31 | 57.39 | 54.32 | 42.83 |
| **B=128** | | | | | | | | | | | | | | | | | |
| SLM | 18.02 | 13.32 | 27.41 | 30.49 | 21.81 | 11.85 | 15.49 | 19.42 | 17.84 | 44.00 | 80.74 | 37.35 | 3.57 | 22.93 | 49.07 | 43.74 | 28.57 |
| Pyramid | 20.78 | 19.76 | 42.92 | 36.31 | 22.37 | 13.91 | 18.84 | 21.84 | 20.42 | 46.50 | **84.55** | 40.23 | 2.79 | 65.46 | 51.49 | 46.83 | 34.69 |
| SnapKV | 20.98 | 19.05 | **44.63** | 35.31 | 22.91 | 13.95 | 18.76 | 21.36 | 20.31 | 45.00 | 83.83 | 39.52 | **3.23** | 64.53 | 52.19 | 47.30 | 34.55 |
| Ada-Pyramid | **21.18** | **20.23** | 44.54 | **37.97** | 22.85 | 14.54 | 19.10 | 21.91 | **20.84** | 52.00 | 84.15 | 39.49 | 2.87 | **71.81** | **52.34** | 47.50 | **35.83** |
| Ada-SnapKV | 20.10 | 20.14 | 44.20 | 37.32 | **23.90** | **15.29** | **19.15** | **22.27** | 20.71 | 50.00 | 84.20 | **39.64** | 3.09 | 67.11 | 52.26 | **48.25** | 35.48 |
| **B=256** | | | | | | | | | | | | | | | | | |
| SLM | 19.08 | 15.30 | 28.27 | 31.87 | 22.26 | 11.37 | 18.08 | 19.37 | 19.99 | 51.00 | 80.92 | 39.62 | 3.57 | 15.90 | 51.74 | 45.22 | 29.60 |
| Pyramid | 20.39 | 22.63 | 46.67 | 38.64 | 23.73 | 15.82 | 21.34 | 22.30 | 22.01 | 57.50 | 83.63 | 40.49 | 2.99 | 75.84 | 53.56 | 50.03 | 37.35 |
| SnapKV | 20.90 | 21.95 | 47.49 | **39.62** | 24.89 | 15.04 | 21.46 | 22.80 | 22.62 | 57.50 | 84.86 | 40.51 | **3.76** | 77.95 | 54.08 | 50.98 | 37.90 |
| Ada-Pyramid | **22.21** | 23.26 | 46.92 | 39.04 | **25.01** | **16.99** | 21.43 | 23.07 | 22.28 | 62.50 | 85.70 | **41.11** | 2.71 | 78.86 | 54.26 | 50.49 | 38.49 |
| Ada-SnapKV | 20.97 | **23.71** | **47.52** | 38.48 | 24.77 | 15.76 | **21.89** | 22.94 | **22.70** | **63.50** | **86.25** | 40.75 | 2.46 | **80.24** | **54.42** | **51.46** | **38.61** |
| **B=512** | | | | | | | | | | | | | | | | | |
| SLM | 21.12 | 15.99 | 30.82 | 30.39 | 22.32 | 10.92 | 21.43 | 19.98 | 22.88 | 61.50 | 82.11 | 41.89 | 3.21 | 17.32 | 53.43 | 47.05 | 31.40 |
| Pyramid | 21.81 | 24.33 | 48.67 | 39.31 | 25.14 | 17.51 | 23.22 | 23.16 | 23.87 | 66.00 | 85.20 | 41.96 | 3.08 | 85.71 | 55.10 | 50.98 | 39.69 |
| SnapKV | **24.29** | **27.83** | **49.00** | 39.63 | 25.26 | 17.63 | **23.53** | 23.36 | 24.44 | 65.00 | 86.18 | 42.01 | **3.27** | 85.29 | 56.10 | 52.73 | 40.35 |
| Ada-Pyramid | 22.92 | 26.37 | 47.66 | 39.49 | 25.52 | **18.50** | 23.43 | **23.53** | 23.87 | 66.50 | 85.42 | 41.98 | 2.59 | 85.49 | 55.14 | 52.29 | 40.04 |
| Ada-SnapKV | 23.62 | 27.34 | 48.70 | **39.81** | **26.42** | 17.36 | 23.42 | 23.26 | **24.50** | **67.50** | **86.46** | **42.04** | 3.04 | **86.64** | **56.11** | **53.05** | **40.58** |
| **B=1024** | | | | | | | | | | | | | | | | | |
| SLM | 22.15 | 18.64 | 31.03 | 32.94 | 22.45 | 11.93 | 23.89 | 20.60 | 25.48 | 64.00 | 84.71 | 41.59 | 3.49 | 22.15 | 53.72 | 49.19 | 33.00 |
| Pyramid | 24.12 | 29.44 | 48.83 | 40.30 | 25.94 | 19.42 | 25.29 | 23.52 | 25.77 | 68.00 | **86.30** | 41.62 | 2.84 | 86.07 | 56.06 | 52.57 | 41.01 |
| SnapKV | 24.10 | **30.11** | 48.97 | 40.97 | 26.89 | 18.12 | **25.93** | 23.75 | 26.02 | 67.50 | 86.25 | 42.33 | 2.94 | 87.23 | **57.07** | 53.43 | 41.35 |
| Ada-Pyramid | 24.82 | 28.94 | 48.47 | **41.44** | 26.58 | **19.75** | 25.08 | **23.84** | 25.54 | 68.00 | 85.80 | 42.44 | **3.51** | 85.68 | 56.45 | 52.72 | 41.22 |
| Ada-SnapKV | **25.11** | 29.98 | **49.27** | 40.62 | **27.05** | 18.54 | 25.85 | 23.46 | **26.08** | **68.50** | 86.30 | 42.77 | 2.92 | **88.27** | 56.88 | **54.16** | **41.61** |
| **B=2048** | | | | | | | | | | | | | | | | | |
| SLM | 22.63 | 22.68 | 35.44 | 33.66 | 22.92 | 13.76 | 26.96 | 20.80 | 26.71 | 66.00 | 85.94 | 42.11 | 2.40 | 25.33 | 57.09 | 52.82 | 34.83 |
| Pyramid | 25.68 | 31.39 | **49.25** | 41.01 | 27.06 | **19.35** | 27.52 | 23.46 | 26.52 | 70.00 | 86.30 | 42.65 | 2.72 | 85.93 | 56.96 | 53.63 | 41.84 |
| SnapKV | 25.79 | 32.54 | 49.18 | 42.09 | 27.31 | 18.91 | 28.50 | 24.01 | 26.83 | 70.00 | **86.46** | 42.86 | **2.95** | 86.56 | 57.24 | 53.89 | 42.20 |
| Ada-Pyramid | 26.41 | 31.74 | 49.08 | 40.95 | **27.79** | 18.89 | 27.54 | 23.88 | 26.78 | **70.50** | 86.30 | **43.44** | 2.61 | 85.93 | 57.22 | 53.56 | 42.04 |
| Ada-SnapKV | **26.60** | **32.68** | 49.10 | **42.65** | 27.27 | 18.97 | **28.60** | **24.17** | **26.86** | 70.00 | 86.30 | 43.12 | 2.45 | **86.81** | **57.33** | **54.11** | **42.31** |

Given $A_i = \text{softmax}(s_i)$ where $s_i = q_i K^T$, we can get $A_i^j = \frac{exp(s_i^j)}{\sum_j exp(s_i^j)}$.

$$\hat{A}_i^j = \mathcal{I}_i^j A_i^j \frac{\sum_j exp(s_i^j)}{\sum_j \mathcal{I}_i^j exp(s_i^j)} = \frac{\mathcal{I}_i^j A_i^j}{||A_i \odot \mathcal{I}_i||_1} \tag{14}$$

Table 7: Detailed results of Llama-3.1-8B-Instruct on LongBench. (Question-agnostic)

| | Single-Doc. QA | | | Multi-Doc. QA | | | Summarization | | | Few-shotLearning | | | Synthetic | | Code | | Ave. Score |
|---|---|---|---|---|---|---|---|---|---|---|---|---|---|---|---|---|---|
| | NrtvQA | Qasper | MF-en | HotpotQA | 2WikiMQA | Musique | GovReport | QMSum | MultiNews | TREC | TriviaQA | SAMSum | PCount | PRe | Lcc | RB-P | |
| Full Cache | 30.22 | 45.37 | 55.80 | 55.97 | 45.00 | 31.26 | 35.12 | 25.38 | 27.20 | 72.50 | 91.64 | 43.57 | 9.41 | 99.50 | 62.88 | 56.43 | 49.20 |
| **B=128** | | | | | | | | | | | | | | | | | |
| SLM | 13.57 | 11.48 | 24.47 | 34.25 | 25.35 | 11.31 | 20.06 | 17.39 | 17.80 | 30.50 | 50.14 | 35.33 | 3.00 | 5.50 | 15.50 | 60.25 | 23.49 |
| Pyramid | **13.91** | 13.05 | **18.71** | 27.43 | 14.36 | 6.35 | 17.93 | 17.25 | 19.52 | 21.00 | 89.09 | 38.03 | 1.11 | 5.00 | 61.93 | 57.00 | 26.35 |
| SnapKV | 12.23 | 11.88 | 17.93 | 25.32 | 12.41 | 8.20 | 17.88 | 16.84 | 19.44 | 22.00 | 90.97 | 37.83 | 2.50 | 5.50 | 61.62 | **57.54** | 26.26 |
| Ada-Pyramid | 13.30 | **13.78** | 17.92 | **31.35** | **15.90** | 7.65 | **19.20** | **17.48** | 19.83 | 24.00 | 90.67 | **38.23** | **4.10** | **6.00** | **63.40** | 56.38 | **27.45** |
| Ada-SnapKV | 13.52 | 12.79 | 18.30 | 28.94 | 14.13 | **9.35** | 19.04 | 17.26 | **19.94** | **25.00** | **91.44** | 37.96 | 1.54 | 4.00 | 63.34 | 56.39 | 27.06 |
| **B=256** | | | | | | | | | | | | | | | | | |
| SLM | 14.25 | 12.92 | 28.75 | 31.80 | 19.19 | 11.39 | 22.86 | 17.26 | 22.73 | 44.00 | 52.80 | 38.77 | 4.00 | 9.00 | 16.11 | 59.79 | 25.35 |
| Pyramid | 13.57 | 17.36 | 19.29 | 30.95 | 18.10 | 8.21 | 20.30 | 17.65 | 20.90 | 26.00 | 90.94 | 40.02 | 4.73 | 4.50 | 64.71 | 55.06 | 28.27 |
| SnapKV | 14.82 | 16.32 | 18.44 | 29.53 | 14.25 | **9.26** | 20.32 | 17.56 | 21.70 | 28.50 | 91.49 | 40.15 | 4.12 | 5.00 | 64.60 | 54.78 | 28.18 |
| Ada-Pyramid | 14.39 | **18.66** | **20.79** | 32.07 | **20.08** | 6.52 | 20.84 | **18.20** | 21.51 | 31.00 | 91.39 | 41.26 | **7.10** | **7.00** | 65.87 | **56.06** | **29.55** |
| Ada-SnapKV | **15.19** | 15.97 | 20.05 | **35.08** | 16.31 | 8.32 | **21.31** | 17.99 | **22.11** | **33.00** | **91.68** | **41.28** | 5.93 | 5.50 | **66.14** | 54.76 | 29.41 |
| **B=512** | | | | | | | | | | | | | | | | | |
| SLM | 14.47 | 16.86 | 34.18 | 31.88 | 18.76 | 11.80 | 26.01 | 18.20 | 25.10 | 53.00 | 60.04 | 41.23 | 4.00 | 9.50 | 19.74 | 57.87 | 27.66 |
| Pyramid | 15.67 | 21.59 | 23.56 | 35.63 | 24.36 | 9.77 | 22.18 | 19.05 | 23.25 | 34.00 | 91.11 | 42.44 | 4.61 | **17.50** | 65.87 | 54.41 | 31.56 |
| SnapKV | 15.22 | 22.90 | 23.41 | 33.00 | 22.54 | 8.91 | 22.63 | 18.31 | 23.50 | 35.00 | 91.39 | 42.83 | 4.50 | 15.50 | **66.79** | 53.75 | 31.20 |
| Ada-Pyramid | 16.05 | **25.46** | **25.75** | 37.04 | **26.72** | **10.65** | 22.80 | **19.75** | 23.28 | 39.50 | 91.65 | 42.64 | 5.75 | 15.00 | 65.55 | **55.73** | 32.71 |
| Ada-SnapKV | **17.72** | 24.22 | 25.38 | **38.00** | 23.47 | 9.25 | **23.55** | 18.91 | **23.82** | **43.00** | **92.14** | **42.81** | **6.34** | 15.00 | 66.12 | 55.32 | **32.82** |
| **B=1024** | | | | | | | | | | | | | | | | | |
| SLM | 13.36 | 21.55 | 41.70 | 32.80 | 22.63 | 12.88 | 28.40 | 19.36 | 25.93 | 62.00 | 75.54 | 41.76 | 2.00 | 11.00 | 22.91 | 57.05 | 30.68 |
| Pyramid | 16.67 | 29.11 | 28.74 | 40.13 | 27.89 | 13.43 | 24.63 | 20.36 | 25.01 | 43.00 | **91.86** | 43.48 | 6.78 | 52.50 | 65.34 | 54.87 | 36.49 |
| SnapKV | 19.57 | **31.67** | 31.52 | **42.04** | 27.89 | 12.94 | 25.31 | 19.66 | 25.12 | 48.50 | 91.37 | 42.83 | 5.99 | **56.00** | **66.56** | 53.97 | 37.56 |
| Ada-Pyramid | **20.04** | 31.37 | **32.77** | 40.91 | **30.40** | 14.35 | 24.97 | **21.36** | 25.37 | 46.50 | 91.61 | 43.78 | **9.13** | 54.50 | 65.35 | 55.27 | 37.98 |
| Ada-SnapKV | 19.74 | 30.47 | 31.42 | 40.82 | 28.98 | **16.07** | **25.35** | 20.57 | **25.57** | **53.50** | 91.76 | **43.81** | 4.63 | 53.50 | 65.75 | **55.90** | **37.99** |
| **B=2048** | | | | | | | | | | | | | | | | | |
| SLM | 16.86 | 31.83 | 48.22 | 33.59 | 29.36 | 15.74 | 30.11 | 20.89 | 26.68 | 65.50 | 92.49 | 44.12 | 5.25 | 21.00 | 39.75 | 56.66 | 36.13 |
| Pyramid | 21.16 | 36.66 | 37.56 | 43.28 | 36.25 | 19.72 | 27.90 | 21.73 | 26.35 | 53.00 | 92.36 | 44.61 | 6.94 | **89.50** | 64.23 | 53.11 | 42.15 |
| SnapKV | 21.24 | 39.86 | 38.21 | 45.96 | 35.36 | **21.20** | **28.68** | 21.49 | 26.50 | 58.00 | **92.36** | 44.06 | 6.31 | 88.50 | **64.26** | 53.73 | 42.86 |
| Ada-Pyramid | 20.75 | 39.12 | **40.18** | 44.49 | **41.17** | 18.94 | 27.90 | **22.06** | 26.71 | 57.50 | 91.86 | 44.35 | **10.30** | 84.50 | 63.72 | **54.81** | 43.02 |
| Ada-SnapKV | **22.42** | **40.26** | 40.13 | **49.60** | 38.23 | 19.70 | 28.49 | 21.76 | **26.77** | **61.50** | 92.35 | 44.04 | 8.27 | 85.00 | 64.11 | 54.09 | **43.55** |

Table 8: Detailed results of Mistral-7B-Instruct-v0.2 on LongBench. (Question-agnostic)

| | Single-Doc. QA | | | Multi-Doc. QA | | | Summarization | | | Few-shotLearning | | | Synthetic | | Code | | Ave. Score |
|---|---|---|---|---|---|---|---|---|---|---|---|---|---|---|---|---|---|
| | NrtvQA | Qasper | MF-en | HotpotQA | 2WikiMQA | Musique | GovReport | QMSum | MultiNews | TREC | TriviaQA | SAMSum | PCount | PRe | Lcc | RB-P | |
| Full Cache | 26.74 | 32.84 | 50.00 | 43.45 | 27.77 | 18.49 | 32.91 | 24.64 | 26.99 | 71.00 | 86.23 | 43.32 | 2.94 | 86.31 | 57.39 | 54.32 | 42.83 |
| **B=128** | | | | | | | | | | | | | | | | | |
| SLM | 7.62 | 7.49 | 26.12 | 18.97 | 14.44 | 7.26 | 19.46 | 17.29 | 19.46 | 26.50 | 73.27 | 36.65 | 1.50 | 4.50 | 14.58 | 59.10 | 22.14 |
| Pyramid | 10.69 | 8.37 | 20.65 | 18.34 | 13.78 | **5.99** | 17.32 | 17.71 | **19.20** | 22.50 | 82.69 | 37.48 | 3.25 | **3.00** | 53.63 | 58.13 | 24.34 |
| SnapKV | 10.51 | 9.52 | 21.28 | 18.76 | 13.85 | 5.57 | 17.68 | 18.05 | 18.18 | 21.50 | 82.52 | 36.61 | **3.74** | 2.50 | 52.84 | 59.04 | 24.51 |
| Ada-Pyramid | **12.53** | **9.61** | 20.94 | 18.54 | 14.38 | 4.90 | 17.54 | **18.15** | 19.02 | 22.00 | **84.44** | **37.72** | 3.67 | 1.33 | **55.49** | 59.28 | 24.97 |
| Ada-SnapKV | 11.13 | 9.33 | **21.78** | **19.41** | **14.96** | 5.95 | **17.82** | 17.39 | 19.15 | **24.50** | 83.71 | 37.33 | 3.73 | 1.50 | 54.91 | **59.92** | **25.16** |
| **B=256** | | | | | | | | | | | | | | | | | |
| SLM | 8.37 | 7.47 | 28.82 | 21.42 | 15.30 | 6.98 | 21.74 | 18.23 | 22.92 | 38.50 | 74.46 | 37.77 | 1.25 | 3.75 | 17.64 | 58.95 | 23.97 |
| Pyramid | 9.86 | **11.24** | 21.86 | 20.71 | 13.40 | 6.58 | 20.07 | 18.21 | 21.04 | 26.00 | 85.73 | 39.34 | 3.11 | **8.68** | 57.18 | 56.31 | 26.21 |
| SnapKV | 10.99 | 10.94 | 23.27 | 21.67 | 13.87 | 6.42 | 20.16 | 18.26 | 21.54 | 28.50 | 85.72 | 39.21 | 2.56 | 6.50 | 57.74 | 58.10 | 26.59 |
| Ada-Pyramid | **11.01** | 9.98 | 22.14 | 22.26 | **15.42** | 7.17 | 19.78 | **18.74** | 21.27 | 31.00 | **87.30** | 39.32 | 3.42 | 8.43 | 58.18 | 58.08 | 27.09 |
| Ada-SnapKV | 10.45 | 10.47 | **23.67** | **22.45** | 13.48 | **7.29** | **20.58** | 18.21 | **21.60** | **32.00** | 87.05 | **39.74** | **3.66** | 8.13 | **58.99** | **58.44** | **27.26** |
| **B=512** | | | | | | | | | | | | | | | | | |
| SLM | 9.54 | 9.05 | 32.27 | 21.64 | 15.17 | 8.76 | 24.89 | 18.33 | 25.39 | 49.00 | 75.51 | 39.40 | 1.50 | 5.75 | 21.43 | 57.64 | 25.95 |
| Pyramid | 11.72 | 12.41 | 23.61 | 23.77 | 14.36 | 6.61 | 21.92 | 19.09 | 23.25 | 37.50 | 87.18 | 40.61 | 3.40 | 14.29 | 59.68 | 57.17 | 28.54 |
| SnapKV | 11.01 | **13.37** | 24.86 | 25.33 | 13.39 | 6.37 | 22.98 | 19.00 | **23.48** | 41.50 | 86.94 | 40.05 | **3.83** | 14.97 | 60.18 | 57.33 | 29.04 |
| Ada-Pyramid | 11.51 | 12.59 | 24.90 | 23.29 | **15.62** | 7.10 | 21.78 | **19.41** | 23.13 | 45.00 | 87.42 | 40.96 | 3.37 | **19.10** | 60.65 | **58.46** | 29.64 |
| Ada-SnapKV | **13.07** | 12.48 | **26.69** | **26.33** | 14.12 | **7.74** | **23.02** | 19.13 | 23.43 | **47.00** | **87.81** | **41.05** | 3.77 | 17.83 | **60.95** | 57.20 | **30.10** |
| **B=1024** | | | | | | | | | | | | | | | | | |
| SLM | 11.05 | 14.15 | 38.89 | 23.79 | 18.25 | 9.71 | 27.20 | 20.25 | 26.20 | 52.50 | 76.96 | 41.79 | 0.00 | 8.50 | 24.24 | 56.58 | 28.13 |
| Pyramid | 12.22 | 16.32 | 28.01 | 26.55 | **15.58** | 7.59 | 23.91 | **20.04** | 25.38 | 46.00 | 87.41 | 41.37 | 3.74 | 34.29 | 63.44 | **58.19** | 31.88 |
| SnapKV | 13.43 | 16.30 | 30.57 | 27.04 | 14.88 | **8.48** | **24.97** | 19.71 | **25.64** | 50.50 | 87.32 | 41.19 | 2.87 | 28.21 | 63.55 | 57.38 | 32.00 |
| Ada-Pyramid | 12.34 | 17.09 | 30.24 | 27.13 | 14.01 | 8.26 | 23.60 | 20.03 | 25.01 | 52.50 | 87.26 | **42.75** | **3.96** | **39.33** | 62.92 | 57.67 | 32.76 |
| Ada-SnapKV | **13.86** | **18.39** | **31.07** | **29.03** | 14.92 | 8.24 | 24.82 | 19.86 | 25.59 | **55.50** | **87.81** | 41.98 | 3.04 | 35.96 | **64.32** | 57.95 | **33.27** |
| **B=2048** | | | | | | | | | | | | | | | | | |
| SLM | 11.80 | 20.40 | 43.35 | 25.01 | 18.21 | 9.80 | 29.40 | 21.40 | 26.73 | 59.00 | 83.06 | 43.26 | 0.38 | 15.75 | 30.11 | 56.05 | 30.86 |
| Pyramid | 14.04 | 21.19 | 36.29 | 27.07 | 15.74 | 10.21 | 25.62 | 20.71 | 26.25 | 52.50 | 87.31 | 43.53 | 4.05 | 65.42 | 64.97 | 58.04 | 35.81 |
| SnapKV | **15.91** | 21.64 | 35.48 | 29.94 | 16.34 | 11.49 | **27.29** | 20.67 | **26.78** | 56.00 | 87.46 | 42.19 | **4.93** | 61.83 | **65.34** | 56.88 | 36.26 |
| Ada-Pyramid | 15.44 | 21.48 | 36.86 | 29.43 | **16.88** | 10.75 | 25.52 | 20.74 | 26.60 | 54.50 | 87.17 | 43.05 | 2.89 | **70.96** | 65.15 | **58.66** | 36.63 |
| Ada-SnapKV | 15.84 | **22.24** | **38.67** | **32.03** | 16.54 | **12.68** | 26.35 | **20.93** | 26.65 | **57.50** | **87.49** | **43.55** | 4.66 | 68.67 | 64.24 | 58.25 | **37.27** |

Then we can obtain:

$$\hat{A}_i = \frac{A_i \odot \mathcal{I}_i}{||A_i \odot \mathcal{I}_i||_1} \tag{15}$$

Thus:

$$\hat{y} = \sum_{i \in [1,h]} \frac{A_i \odot \mathcal{I}_i}{||A_i \odot \mathcal{I}_i||_1} V_i W_i^O \tag{16}$$

By calculating the L1 distance between their outputs, we can obtain

$$||y - \hat{y}||_1 = || \sum_{i \in [1,h]} (\mathbf{1} - \frac{\mathcal{I}_i}{||A_i \odot \mathcal{I}_i||_1}) \odot A_i V_i W_i^O ||_1 \tag{17}$$

$$\leq \sum_{i \in [1,h]} ||(\mathbf{1} - \frac{\mathcal{I}_i}{||A_i \odot \mathcal{I}_i||_1}) \odot A_i V_i W_i^O ||_1 \tag{18}$$

$$\leq \sum_{i \in [1,h]} ||(\mathbf{1} - \frac{\mathcal{I}_i}{||A_i \odot \mathcal{I}_i||_1}) \odot A_i ||_1 ||V_i W_i^O ||_\infty \tag{19}$$

$$\leq C \sum_{i \in [1,h]} ||(\mathbf{1} - \frac{\mathcal{I}_i}{||A_i \odot \mathcal{I}_i||_1}) \odot A_i ||_1 \tag{20}$$

$$where\ C = Max\left\{ ||V_i W_i^O ||_\infty \right\}$$

By expanding $A_i$, we can further simplify the expression.

$$Let\ ||A_i \odot \mathcal{I}_i||_1\ as\ F_i \in (0,1) \tag{21}$$

$$||y - \hat{y}||_1 \leq C \sum_{i \in [1,h]} ||(\mathbf{1} - \frac{\mathcal{I}_i}{||A_i \odot \mathcal{I}_i||_1}) \odot A_i ||_1 \tag{22}$$

$$= C \sum_{i \in [1,h]} \sum_{j \in [1,n]} \frac{|F_i - \mathcal{I}_i^j| A_i^j}{F_i} \tag{23}$$

Considering

$$\mathcal{I}_i^j = \begin{cases} 1 & \text{if } K_i^j \text{ and } V_i^j \text{ are retained} \\ 0 & \text{otherwise, evict } K_i^j \text{ and } V_i^j \end{cases} \text{ and } \sum_{j \in [1,n]} A_i^j = 1 \tag{24}$$

$$||y - \hat{y}||_1 \leq C \sum_{i \in [1,h]} \overset{if \mathcal{I}_i^j = 0}{\sum_{j \in [1,n]}} A_i^j + C \sum_{i \in [1,h]} \frac{(1 - F_i)}{F_i} \overset{if \mathcal{I}_i^j = 1}{\sum_{j \in [1,n]}} A_i^j \tag{25}$$

Due to $F_i = ||A_i \odot \mathcal{I}_i||_1 = \sum_{j \in [1,n]} \mathcal{I}_i^j A_i^j = \overset{if \mathcal{I}_i^j = 1}{\sum_{j \in [1,n]}} A_i^j$

$$||y - \hat{y}||_1 \leq C \sum_{i \in [1,h]} \overset{if \mathcal{I}_i^j = 0}{\sum_{j \in [1,n]}} A_i^j + C \sum_{i \in [1,h]} (1 - \overset{if \mathcal{I}_i^j = 1}{\sum_{j \in [1,n]}} A_i^j) \tag{26}$$

$$= 2C \sum_{i \in [1,h]} \overset{if \mathcal{I}_i^j = 0}{\sum_{j \in [1,n]}} A_i^j \tag{27}$$

$$= 2C \sum_{i \in [1,h]} \sum_{j \in [1,n]} (1 - \mathcal{I}_i^j) A_i^j \tag{28}$$

$$= 2hC - 2C \sum_{i \in [1,h]} \sum_{j \in [1,n]} \mathcal{I}_i^j A_i^j \tag{29}$$

Finally,

$$L_1 \text{ Eviction Loss} \leq \epsilon = 2hC - 2C \sum_{i \in [1,h]} \sum_{j \in [1,n]} \mathcal{I}_i^j A_i^j \tag{30}$$

$$\square$$

## A.9 Proof of Theorem 3.2

**Theorem A.2.** *The Top-k cache eviction $\{\mathcal{I}_i^*\}$ minimizes the upper bound $\epsilon$ of $L_1$ eviction loss:*

$$\text{Top-k eviction decision } \{\mathcal{I}_i^*\} = \underset{\{\mathcal{I}_i\}}{\arg\min}\, \epsilon\,, \tag{31}$$

$$\epsilon^* = \min_{\{\mathcal{I}_i\}} \epsilon = 2hC - 2C \sum_{i \in [1,h]} \overset{A_i^j \in \text{Top-k}(A_i, B_i)}{\sum_{j \in [1,n]}} A_i^j \tag{32}$$

*Proof.* Given the budget allocation results $\{B_i\}$, the objective of minimizing $\epsilon$ can be decomposed into $i \in [1, h]$ independent subproblems, each expressed as follows:

$$\underset{\mathcal{I}_i}{\arg\max} \sum_{j \in [1,n]} \mathcal{I}_i^j A_i^j, \quad \text{s.t.} \quad \sum_{j \in [1,n]} \mathcal{I}_i^j = B_i \tag{33}$$

The Top-k cache eviction maximizes the $h$ independent subproblems, thereby minimizing the upper bound $\epsilon$. Formally:

$$\mathcal{I}_i^* = \underset{\mathcal{I}_i}{\arg\max} \sum_{j \in [1,n]} \mathcal{I}_i^j A_i^j \tag{34}$$

The optimal solution satisfies:

$$\min_{\mathcal{I}_i} \sum_{j \in [1,n]} \mathcal{I}_i^j A_i^j = \sum_{j \in [1,n]} \mathcal{I}_i^{*j} A_i^j = \overset{A_i^j \in \text{Top-k}(A_i, B_i)}{\sum_{j \in [1,n]}} A_i^j \tag{35}$$

$\square$

## A.10 Proof of Theorem 3.3

**Theorem A.3.** *The adaptive budget allocation $\{B_i^*\}$ derived from Algorithm 1 achieves the minimal upper bound $\epsilon^{**}$ for loss associated with Top-k eviction methods:*

$$\epsilon^{**} = \min_{\{B_i\}} \epsilon^* \tag{36}$$

*Proof.* From line 3 of Algorithm 1, which adaptively allocates budgets $\{B_i^*\}$ based on the Top-k indices $T = \text{Top-k}(A, B)$.indices, it follows that the resulting Top-k eviction leads to an upper bound $\epsilon^{**}$:

$$\epsilon^{**} = 2hC - 2C \sum_{i \in [1,h]} \overset{A_i^j \in \text{Top-k}(A_i, B_i^*)}{\sum_{j \in [1,n]}} A_i^j \tag{37}$$

$$= 2hC - 2C \sum_{i \in [1,h]} \overset{A_i^j \in \text{Top-k}(A, B)}{\sum_{j \in [1,n]}} A_i^j \tag{38}$$

Given a fixed overall budget, i.e., $\sum_{i \in [1,h]} B_i = \sum_{i \in [1,h]} B_i^*$, we can derive that $\epsilon^{**} \leq \epsilon^*$ for any budget allocation results $\{B_i\}$. This is because, in the upper bound, the term $\sum_{i \in [1,h]} \sum_{j \in [1,n]}^{A_i^j \in \text{Top-k}(A,B)} A_i^j \geq \sum_{i \in [1,h]} \sum_{j \in [1,n]}^{A_i^j \in \text{Top-k}(A_i, B_i)} A_i^j$. This result can also be understood as a global optimal solution always outperforming a local optimal solution. $\square$

## A.11 Detailed Visualization of Head Concentration

Figure 11 supplements Figure 1a in the main paper by presenting the visualization results across all layers. It can be observed that in all layers, different heads exhibit significant variations in attention concentration. This indicates that the adaptive allocation strategy has great potential to reduce the eviction loss in practice.

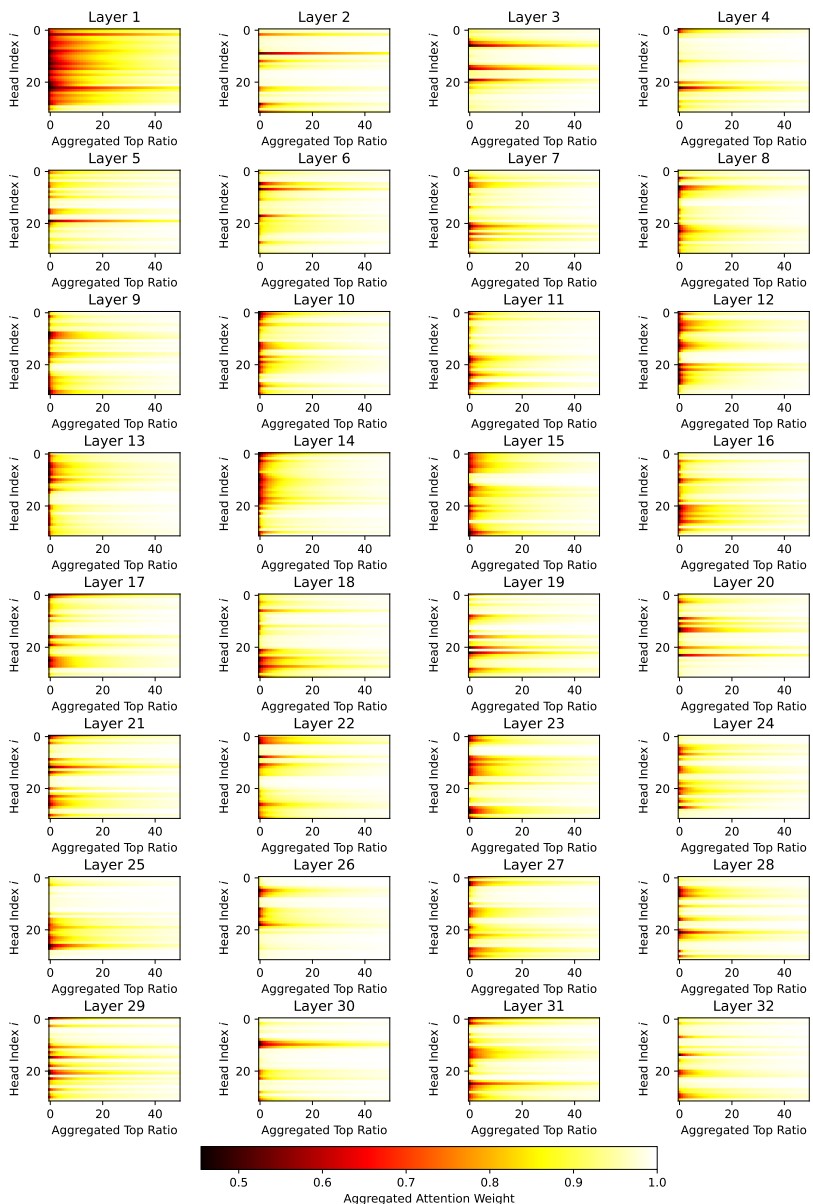

Figure 11: Visualization of Heads' Concentrations

Table 9: S-NIAH and MK-NIAH templates in Ruler Benchmark. [32]

| | |
|---|---|
| Single NIAH Subtask-1 (S-NIAH-1) | **Task Template:**
Some special magic numbers are hidden within the following text. Make sure to memorize it. I will quiz you about the numbers afterwards.
The grass is green. The sky is blue. The sun is yellow. Here we go. There and back again.
...... One of the special magic numbers for {word} is: {number}. ......
What is the special magic number for {word} mentioned in the provided text?

The special magic number for {word} mentioned in the provided text is |
| Single NIAH Subtask-2 (S-NIAH-2) | **Task Template:**
Some special magic numbers are hidden within the following text. Make sure to memorize it. I will quiz you about the numbers afterwards.
Paul Graham Essays.
...... One of the special magic numbers for {word} is: {number}. ......
What is the special magic number for {word} mentioned in the provided text?

The special magic number for {word} mentioned in the provided text is |
| Single NIAH Subtask-3 (S-NIAH-3) | **Task Template:**
Some special magic words are hidden within the following text. Make sure to memorize it. I will quiz you about the words afterwards.
Paul Graham Essays.
...... One of the special magic words for {word} is: {word}. ......
What is the special magic word for {word} mentioned in the provided text?

The special magic word for {word} mentioned in the provided text is |
| Multi-keys NIAH Subtask-1 (MK-NIAH-1) | **Task Template:**
Some special magic numbers are hidden within the following text. Make sure to memorize it. I will quiz you about the numbers afterwards.
Paul Graham Essays.
...... One of the special magic numbers for {word-1} is: {number-1}. ......
...... One of the special magic numbers for {word-2} is: {number-2}. ......
...... One of the special magic numbers for {word-3} is: {number-3}. ......
...... One of the special magic numbers for {word-4} is: {number-4}. ......
What is the special magic number for {word-4} mentioned in the provided text?

The special magic number for {word-4} mentioned in the provided text is |
| Multi-keys NIAH Subtask-2 (MK-NIAH-2) | **Task Template:**
Some special magic numbers are hidden within the following text. Make sure to memorize it. I will quiz you about the numbers afterwards.
One of the special magic numbers for {word-1} is: {number-1}.
One of the special magic numbers for {word-2} is: {number-2}.
...... One of the special magic numbers for {word-x} is: {number-x}. ......
One of the special magic numbers for {word-n-1} is: {number-n-1}.
One of the special magic numbers for {word-n} is: {number-n}.
What is the special magic number for {word-x} mentioned in the provided text?

The special magic number for {word-x} mentioned in the provided text is |
| Multi-keys NIAH Subtask-3 (MK-NIAH-3) | **Task Template:**
Some special magic uuids are hidden within the following text. Make sure to memorize it. I will quiz you about the uuids afterwards.
One of the special magic uuids for {uuid-1} is: {uuid-1}.
One of the special magic uuids for {uuid-2} is: {uuid-2}.
...... One of the special magic uuids for {uuid-x} is: {uuid-x}. ......
One of the special magic uuids for {uuid-n-1} is: {uuid-n-1}.
One of the special magic uuids for {uuid-n} is: {uuid-n}.
What is the special magic number for {uuid-x} mentioned in the provided text?

The special magic number for {uuid-x} mentioned in the provided text is |

Table 10: MV-NIAH, MQ-NIAH, VT, CWE, and FWE templates in Ruler Benchmark. [32]

| | |
|---|---|
| Multi-values NIAH (MV-NIAH) | **Task Template:**
Some special magic numbers are hidden within the following text. Make sure to memorize it. I will quiz you about the numbers afterwards.
Paul Graham Essays.
...... One of the special magic numbers for {word} is: {number-1}. ......
...... One of the special magic numbers for {word} is: {number-2}. ......
...... One of the special magic numbers for {word} is: {number-3}. ......
...... One of the special magic numbers for {word} is: {number-4}. ......
What are all the special magic numbers for {word} mentioned in the provided text?

The special magic numbers for {word} mentioned in the provided text are |
| Multi-queries NIAH (MQ-NIAH) | **Task Template:**
Some special magic numbers are hidden within the following text. Make sure to memorize it. I will quiz you about the numbers afterwards.
Paul Graham Essays.
...... One of the special magic numbers for {word-1} is: {number-1}. ......
...... One of the special magic numbers for {word-2} is: {number-2}. ......
...... One of the special magic numbers for {word-3} is: {number-3}. ......
...... One of the special magic numbers for {word-4} is: {number-4}. ......
What are all the special magic numbers for {word-1}, {word-2}, {word-3}, and {word-4} mentioned in the provided text?

The special magic numbers for {word-1}, {word-2}, {word-3}, and {word-4} mentioned in the provided text are |
| Variable Tracking (VT) | **Task Template:**
{one task example}
Memorize and track the chain(s) of variable assignment hidden in the following text.

The grass is green. The sky is blue. The sun is yellow. Here we go. There and back again. ...... VAR {X1} = {number} ......
...... VAR {X2} = {X1} ......
...... VAR {X3} = {X2} ......
...... VAR {X4} = {X3} ......
...... VAR {X5} = {X4} ......
Question: Find all variables that are assigned the value {number} in the text above.

Answer: According to the chain(s) of variable assignment in the text above, 5 variables are assigned the value {number}, they are: |
| Common Words Extraction (CWE) | **Task Template:**
{one task example}
Below is a numbered list of words. In these words, some appear more often than others. Memorize the ones that appear most often.
1. word-a 2. word-b 3. word-c 4. word-a 5. word-d 6. word-a 7. word-e 8. word-f ......
Question: What are the 10 most common words in the above list?

Answer: The top 10 words that appear most often in the list are: |
| Frequent Words Extraction (FWE) | **Task Template:**
Read the following coded text and track the frequency of each coded word. Find the three most frequently appeared coded words. ... ... word-a ... word-b ... ... ... word-c ... word-a ... word-d word-e ... word-a ... ... word-f ... ... ... word-g ... word-h ... word-a ... word-i ......
Question: Do not provide any explanation. Please ignore the dots '....'. What are the three most frequently appeared words in the above coded text?

Answer: According to the coded text above, the three most frequently appeared words are: |

Table 11: QA templates in Ruler Benchmark. [32]

| | |
|---|---|
| Single Hop QA (S-QA) | **Task Template:**
Answer the question based on the given documents. Only give me the answer and do not output any other words.

The following are given documents.

Document 1:
{document-1}
......
Document x:
{document-x}
......
Document n:
{document-n}

Answer the question based on the given documents. Only give me the answer and do not output any other words.

Question: {question}

Answer: |
| Multi Hop QA (M-QA) | **Task Template:**
Answer the question based on the given documents. Only give me the answer and do not output any other words.

The following are given documents.

Document 1:
{document-1}
......
Document x:
{document-x}
......
Document y:
{document-y}
......
Document n:
{document-n}

Answer the question based on the given documents. Only give me the answer and do not output any other words.

Question: {question}

Answer: |

Table 12: Details of 16 Datasets in LongBench

| Label | Task | Task Type | Eval metric | Avg len | Language | Sample Num |
|-------|------|-----------|-------------|---------|----------|------------|
| NrtvQA | NarrativeQA | Single-Doc. QA | F1 | 18,409 | EN | 200 |
| Qasper | Qasper | Single-Doc. QA | F1 | 3,619 | EN | 200 |
| MF-en | MultiFieldQA-en | Single-Doc. QA | F1 | 4,559 | EN | 150 |
| HotpotQA | HotpotQA | Multi-Doc. QA | F1 | 9,151 | EN | 200 |
| 2WikiMQA | 2WikiMultihopQA | Multi-Doc. QA | F1 | 4,887 | EN | 200 |
| Musique | MuSiQue | Multi-Doc. QA | F1 | 11,214 | EN | 200 |
| GovReport | GovReport | Summarization | Rouge-L | 8,734 | EN | 200 |
| QMSum | QMSum | Summarization | Rouge-L | 10,614 | EN | 200 |
| MultiNews | MultiNews | Summarization | Rouge-L | 2,113 | EN | 200 |
| TREC | TREC | Few-shotLearning | Accuracy | 5,177 | EN | 200 |
| TriviaQA | TriviaQA | Few-shotLearning | F1 | 8,209 | EN | 200 |
| SAMSum | SAMSum | Few-shotLearning | Rouge-L | 6,258 | EN | 200 |
| PCount | PassageCount | Synthetic | Accuracy | 11,141 | EN | 200 |
| PRe | PassageRetrieval-en | Synthetic | Accuracy | 9,289 | EN | 200 |
| Lcc | LCC | Code | Edit Sim | 1,235 | Python/C#/Java | 500 |
| RB-P | RepoBench-P | Code | Edit Sim | 4,206 | Python/Java | 500 |

Table 13: LongBench templates. Single-Doc. QA Tasks.

| | |
|---|---|
| NarrativeQA | **Task Template:** You are given a story, which can be either a novel or a movie script, and a question. Answer the question asconcisely as you can, using a single phrase if possible. Do not provide any explanation. 

 Story: {context} 

 Now, answer the question based on the story asconcisely as you can, using a single phrase if possible. Do not provide any explanation. 

 Question: {question} |
| Qasper | **Task Template:** You are given a scientific article and a question. Answer the question as concisely as you can, using a single phrase or sentence if possible. If the question cannot be answered based on the information in the article, write "unanswerable". If the question is a yes/no question, answer "yes", "no", or "unanswerable". Do not provide any explanation. 

 Article: {context} 

 Answer the question based on the above article as concisely as you can, using a single phrase or sentence if possible. If the question cannot be answered based on the information in the article, write "unanswerable". If the question is a yes/no question, answer "yes", "no", or "unanswerable". Do not provide any explanation. 

 Question: {question} |
| MultifieldQA EN | **Task Template:** Read the following text and answer briefly. 

 {context} 

 Now, answer the following question based on the above text, only give me the answer and do not output any other words. 

 Question: {question} |

Table 14: LongBench templates. Multi-Doc. QA Tasks.

| | |
|---|---|
| HotpotQA | **Task Template:**
Answer the question based on the given passages. Only give me the answer and do not output any other words.

The following are given passages.
{context}

Answer the question based on the given passages. Only give me the answer and do not output any other words.

Question: {question} |
| 2WikimQA | **Task Template:**
Answer the question based on the given passages. Only give me the answer and do not output any other words.

The following are given passages.
{context}

Answer the question based on the given passages. Only give me the answer and do not output any other words.

Question: {question} |
| Musique | **Task Template:**
Answer the question based on the given passages. Only give me the answer and do not output any other words.

The following are given passages.
{context}

Answer the question based on the given passages. Only give me the answer and do not output any other words.

Question: {question} |

Table 15: LongBench templates. Summarization Tasks.

| | |
|---|---|
| Gov Report | **Task Template:**
You are given a report by a government agency. Write a one-page summary of the report.

Report:
{context}

Now, write a one-page summary of the report. |
| QMSum | **Task Template:**
You are given a meeting transcript and a query containing a question or instruction. Answer the query in one or more sentences.

Transcript:
{context}

Now, answer the query based on the above meeting transcript in one or more sentences.

Query: {question} |
| Multi News | **Task Template:**
You are given several news passages. Write a one-page summary of all news.

News:
{context}

Now, write a one-page summary of all the news. |

Table 16: LongBench templates. Few-shot Learning Tasks.

| | |
|---|---|
| TREC | **Task Template:**
Please determine the type of the question below. Here are some examples of questions.

{context}
{question} |
| TriviaQA | **Task Template:**
Answer the question based on the given passage. Only give me the answer and do not output any other words. The following are some examples.

{context}

{question} |
| SAMSum | **Task Template:**
Summarize the dialogue into a few short sentences. The following are some examples.

{context}

{question} |

Table 17: LongBench templates. Synthetic Tasks.

| | |
|---|---|
| Passage Count | **Task Template:**
There are some paragraphs below sourced from Wikipedia. Some of them may be duplicates. Please carefully read these paragraphs and determine how many unique paragraphs there are after removing duplicates. In other words, how many non-repeating paragraphs are there in total?
{context}

Please enter the final count of unique paragraphs after removing duplicates. The output format should only contain the number, such as 1, 2, 3, and so on. |
| Passage Retrieval EN | **Task Template:**
Here are 30 paragraphs from Wikipedia, along with an abstract. Please determine which paragraph the abstract is from.

{context}

The following is an abstract.

{question}

Please enter the number of the paragraph that the abstract is from. The answer format must be like "Paragraph 1", "Paragraph 2", etc. |

Table 18: LongBench templates. Code Tasks.

| | |
|---|---|
| Lcc | **Task Template:**
Please complete the code given below.
{context}
Next line of code: |
| Repobench-P | **Task Template:**
Please complete the code given below.
{context}
{question}
Next line of code: |

