# OpenReview forum: "Ada-KV: Optimizing KV Cache Eviction by Adaptive Budget Allocation for Efficient LLM Inference"
_NeurIPS.cc/2025/Conference — NeurIPS 2025 poster_

### Official Review · Reviewer_1zqk · 2025-06-04

**Clarity:** 3
**Significance:** 2
**Originality:** 2
**Rating:** 2
**Confidence:** 5

**Summary:**

The paper addresses efficiency bottlenecks in LLM inference caused by the size of key-value caches, particularly under long context scenarios. Existing cache eviction strategies typically allocate compression budgets uniformly across attention heads, which fails to exploit varying head-wise attention sparsity. The authors propose Ada-KV, a novel head-wise adaptive budget allocation strategy that minimizes a derived upper bound on attention output loss. Ada-KV is shown to be plug-and-play with prior Top-k based eviction schemes (e.g., SnapKV, Pyramid). Extensive evaluations on Ruler and LongBench benchmarks in both question-aware and question-agnostic settings show consistent improvements in generation quality while preserving computational efficiency.

**Questions:**

1. Some very related works [1,2,3] are overlooked. The authors also need to explain the difference between the proposed method and these works in detail.
2. After carefully reviewing Section 3.2 and Appendix A.8, I believe the theoretical component is not essential to the core contributions of the paper. Prior works [1,2,3] have already established that attention heads exhibit varying degrees of sparsity. Given this, it is rather intuitive that assigning the same budget to heads with different sparsity levels is suboptimal. In my view, the introduction of Section 3.2 does not significantly strengthen the paper.
3. It is unclear whether the head-wise budget sizes are dynamically adjusted during the decoding phase. Based on the current description, they appear to be fixed. However, it is possible that the optimal budget sizes could evolve as new tokens are generated. This raises the question of whether the budgets are determined by the model itself or influenced by the input text. If the latter is true, further analysis would be necessary to understand this dynamic behaviour. Compared with the current theoretical component, I also believe this question warrants a more formal theoretical analysis.
4. Although the idea of allocating different budget sizes for different heads is interesting and has been proved by lots of previous papers, I do not observe substantial improvements over the baseline methods to which Ada-KV is applied (around 1% improvement on average). Could the authors explain why?
5. When the budget size is small (10% of the original cache size) the performance degradation is huge compared with Full Cache (37.45% vs. 49.20%, as presented in Table 1). Given that the current KV-cache optimization approaches [4,5,6] can achieve much better performance with even smaller budget sizes (5% or less), I suggest applying Ada-KV to other base models.


Reference:
1. Minference: https://arxiv.org/pdf/2407.02490
2. HeadKV: https://arxiv.org/pdf/2410.19258
3. DuoAttention: https://arxiv.org/pdf/2410.10819
4. ArkVale: https://openreview.net/pdf?id=4oAt5L4lYe
5. SparQ: https://arxiv.org/pdf/2312.04985
6. PQCache: https://arxiv.org/pdf/2407.12820

**Ethical Concerns:**

["NO or VERY MINOR ethics concerns only"]

**Final Justification:**

I appreciate the authors’ reply. In their response, they highlight the distinction between eviction-based approaches and dynamic sparsity methods. My main concern is that eviction-based methods were widely used in prior work primarily because, at the time, dynamic approaches were considered costly due to retrieval overhead and GPU–CPU offloading. However, recent research has shown that many dynamic approaches are both efficient and capable of achieving much better approximations, as I noted in Question 5. Ada-KV could be seen as an inspiring contribution in 2024, but given the current state of the field, I believe that for the work to be accepted, it should demonstrate its efficacy when integrated with modern dynamic sparsity methods such as Arkvale and PQCache. Unfortunately, even though I already suggested it in my review, the authors did not provide such evidence in either the main paper or the rebuttal. Therefore, I will maintain my current score.

**Limitations:**

The paper does not cover the limitations.

**Quality:**

2

**Strengths And Weaknesses:**

Strengths:
1. The paper addresses a critical and timely challenge in long-context LLM inference;
2. Two benchmarks (LongBench and Ruler) are tested in the experiments.

Weaknesses:
1. The paper overlooks several relevant prior works that also exploit per-head or per-group cache eviction;
2. The performance of the proposed method is not convincing.

More details can be found in [Questions].

---

> ### Author Rebuttal · Authors · 2025-07-31
>
> # Reviewer 1zqk
>
> Thank you for your time and effort in reviewing our work. We address your concerns below and are happy to discuss any further questions.
>
> ## Reply to W1&Q1:
>
> > Some very related works [1,2,3] are overlooked. The authors also need to explain the difference between the proposed method and these works in detail.
>
> **About MInference [1] and Dynamic Sparse Attention Methods**
>
> MInference focuses on offline analysis of head patterns for prefilling acceleration. Along with methods such as [4,5,6], it belongs to the **category of dynamic sparse attention**. These methods differ fundamentally from the **cache eviction** methods we study. Specifically, dynamic sparse attention methods retain the full KV cache but dynamically select subsets for computation at each step, typically to reduce computation. In contrast, cache eviction removes less important KV entries to reduce memory usage directly. Please see our response to Q5 for more detailed discussion. We will incorporate these distinctions in the revised Related Work section.
>
> **About HeadKV [2] and DuoAttention [3]**
>
> Both HeadKV and DuoAttention rely on **offline**, **training-based** analysis of attention heads to evaluate cache importance. As noted in the DuoAttention paper, **their method requires significant computational resources** (e.g., 8×NVIDIA A100 GPUs) for training with 7–8B parameter models, and no results are reported at the 70B scale—likely due to the prohibitive cost. In contrast, **our method is plug-and-play**, requiring no additional training cost.
>
> Functionally, DuoAttention does not support adaptive allocation—it only converts certain heads into sliding window style. Notably, HeadKV, on the other hand, uses the Ada-KV's CUDA kernel and code to implement its adaptive head-wise budget allocation after its offline training.
>
> **On General Applicability**
>
> Ada-KV is designed as a plug-and-play enhancement that can be easily integrated into existing cache eviction methods. In our paper, we demonstrate this by applying it to the widely recognized SOTA methods such as SnapKV and PyramidKV. Additionally, several new methods were released publicly after our submission (to maintain double-blind review, we refer to them as Anonymous Methods; links will be provided after publication). We found that Ada-KV could deliver substantial improvements on these concurrent methods, outperforming even training-based approaches such as DuoAttention and HeadKV. Below, we present an example result achieved by enhancing a method with Ada-KV. We will include these new results as case studies of Ada-KV's effectiveness in the Appendix of the revised version.
>
> | Question-Agnostic LongBench Ave. Score | 20% Cache | 40% Cache |
> |---|---|---|
> | DuoAttention (Training-based) | 39.52 | 48.17 |
> | HeadKV (Training-based) | 42.64 | 47.23 |
> | SnapKV  | 40.21 | 44.92 |
> | SnapKV + Ada-KV (Ada-SnapKV) | 41.39 | 45.74 |
> | Anonymous Method | 43.78 | 47.76 |
> | **Anonymous Method + Ada-KV** | **46.68** | **49.21** |
>
>
> ## Reply to W2 & Q4:
>
> > ... I do not observe substantial improvements over the baseline methods to which Ada-KV is applied (around 1% improvement on average). ...
>
> First, the improvements brought by Ada-KV on the Ruler benchmark—specifically designed for long-context memory capacity of LLMs—are substantial and should not be overlooked. For example, **in the challenging question-agnostic setting, Ada-KV improves SnapKV from 78.08 to 88.17 at 60% cache size.** On average, this results in a 9-point improvement across different cache sizes.
>
> Regarding LongBench, it covers six task types across sixteen datasets. As the task-level analysis in Section 4.3 (line 303) of our paper, certain tasks, especially the Code task, are largely insensitive to cache compression. As shown in Table 2, the Code task incurs no loss even at 20% cache budget, indicating minimal room for further improvement. In contrast, Ada-KV increases performance from 32.54 to 36.05 on Single-Doc QA task. This suggests that performance gains are correlated with the difficulty of long-context task. This also explains why improvements are much more significant on challenging long-context benchmarks Ruler.
>
> Finally, it is worth emphasizing that Ada-KV is a plug-and-play enhancement with no additional training cost. Its consistent improvements across existing and emerging methods (see Reply Q1: On General Applicability) demonstrate broad applicability that goes beyond average percentage gains.
>
>
> ## Reply to Q2:
>
> > ... it is rather intuitive that assigning the same budget to heads with different sparsity levels is suboptimal. In my view, the introduction of Section 3.2 does not significantly strengthen the paper.
>
> We appreciate your suggestion and will revise Section 3.2 to be more concise, so as to reallocate space to present more experimental results in the main text.
>
> That said, while the intuition behind head-wise budget allocation is straightforward, we believe that a formal theoretical analysis still adds significant value. Our theoretical analysis for the first time formally provides an upper bound on the loss under head-wise budget allocation, which offers a sound justification for the design of Ada-KV. This not only enhances the interpretability of our method but also lays a foundation for future research.
>
>
>
>
> ## Reply to Q3:
>
> > It is unclear whether the head-wise budget sizes are dynamically adjusted during the decoding phase. Based on the current description, they appear to be fixed. However, it is possible that the optimal budget sizes could evolve as new tokens are generated. This raises the question of whether the budgets are determined by the model itself or influenced by the input text. If the latter is true, further analysis would be necessary to understand this dynamic behaviour. Compared with the current theoretical component, I also believe this question warrants a more formal theoretical analysis.
>
> We clarify that Ada-KV follows the standard practice of previous methods (SnapKV and PyramidKV): cache eviction is applied once after the prefilling phase at each layer. No further cache eviction is performed during decoding, so the budget remains fixed.
>
> We agree that optimal budgets may depend on both the model and the input, and could shift during generation. However, in cache eviction settings, dynamically adjusting budgets is difficult—once entries are evicted, they cannot be recovered. In contrast, dynamic sparse attention retains the full KV cache and allows for such adaptability. While this makes dynamic budget adjustment feasible in that context, it lies outside the scope of our work (see response to Q1). We see this as an interesting direction for future research and discuss further the connection to dynamic sparse attention (see response to Q5).
>
>
>
> ## Reply to Q5:
>
> > ...Given that the current KV-cache optimization approaches [4,5,6] can achieve much better performance with even smaller budget sizes (5% or less), I suggest applying Ada-KV to other base models.
>
>
> First, we would like to clarify that the three referenced papers [4, 5, 6] fall under the category of dynamic sparse attention, which differs fundamentally from the cache eviction studied in our work. **So, a direct comparison is not appropriate due to the different definitions of "budget".**
>
>
> Dynamic sparse attention [4, 5, 6] retains the **entire** KV cache in memory, selecting only a portion (the "budget") per step for computation. While this approach reduces computation, it does **not** reduce memory usage—regardless of whether the full KV cache is stored on the CPU [4, 6] or GPU [5]. Moreover, if stored on the CPU, retrieval indexing overhead and PCIe bandwidth quickly become bottlenecks. In contrast, cache eviction permanently removes less important KV entries, reducing both memory consumption and attention computation.
>
>
> Because of their complementary strengths and differing goals, dynamic sparsity and cache eviction are viewed as **orthogonal** approaches, which can be combined in practice. For example, one can first apply cache eviction to reduce the KV cache to a moderate size (e.g., 50%), then apply dynamic sparsity (e.g., using only 5% of the retained cache per step) to further reduce computation cost. We believe exploring such hybrid strategies is a promising direction for future work.
>
>
> [1] Minference 1.0: Accelerating pre-filling for long-context llms via dynamic sparse attention
>
> [2] HeadKV: Not all heads matter: A head-level kv cache compression method with integrated retrieval and reasoning
>
> [3] DuoAttention: Efficient long-context llm inference with retrieval and streaming heads
>
> [4] ArkVale: Efficient generative llm inference with recallable key-value eviction
>
> [5] SparQ attention: Bandwidth-efficient llm inference
>
> [6] PQCache: Product quantization-based kvcache for long context llm inference

---

> ### Author Response · Authors · 2025-08-05
> **Kind Request for Your Feedback on Our Clarifications**
>
> Dear Reviewer 1zqk,
>
> Thank you very much for your constructive feedback of our manuscript. We have provided detailed responses to each of your comments in order to address your concerns.
>
> As the discussion period draws to a close, we are eager to hear whether our responses have addressed your concerns. If you have any further concerns or require additional clarification, please let us know—we would be happy to provide more details.
>
> We sincerely hope our responses have resolved your concerns and would be grateful if you could take them into consideration in your final evaluation.
>
> Thank you again for your time and thoughtful input.
>
> Best regards,
>
> The Authors

---

> > ### Comment · Reviewer_1zqk · 2025-08-07
> >
> > I appreciate the authors’ reply. My main concern is that eviction-based methods were widely used in prior work primarily because, at the time, dynamic approaches were considered costly due to retrieval overhead and GPU–CPU offloading. However, recent research has shown that many dynamic approaches are both efficient and capable of achieving much better approximations, as I noted in Question 5. Ada-KV could be seen as an inspiring contribution in 2024, but given the current state of the field, I believe that for the work to be accepted, it should demonstrate its efficacy when integrated with modern dynamic sparsity methods such as Arkvale and PQCache. Therefore, I will maintain my current score.

---

> ### Author Response · Authors · 2025-08-07
>
> Thank you for your response and for recognizing Ada-KV as an inspiring contribution to eviction-based area.
>
> **However, we respectfully disagree that eviction-based methods lost value due to advances in offloading dynamic sparsity.  These are fundamentally orthogonal research directions that can be combined in practice.** Cache eviction remains an active area, with many new works continually emerging, as we mentioned in our Reply to Q1. These ongoing developments underscore the importance and attention  of this research direction, so it would be inappropriate to simply dismiss these works based on orthogonal research.
>
> **It is important to note that "budget" has different meanings depending on the research area: for dynamic sparsity with offloading, it refers to the portion of KV cache recalled from CPU to GPU during decoding steps, while for cache eviction, it refers to the fraction of KV cache retained in  memory after compression.** Therefore, a direct comparison of their respective "budgets" is inappropriate. Let’s take another orthogonal technique—KV cache quantization—as an example. It can often reduce the KV cache “budget” to 50% or even 25% (e.g., converting from bf16 to INT8 or INT4), while dynamic sparsity may achieve smaller "budgets" (e.g., 5% or less), as you mentioned. However, it would clearly be inappropriate to dismiss the value of KV cache quantization based on such a comparison.
>
> **Therefore, we believe that cache eviction remains a highly valuable research area, and Ada-KV is an inspiring contribution that helps advance this field.** We also agreee that applying Ada-KV-like ideas to dynamic sparsity is indeed a promising direction. However, this lies beyond the scope of our current work, and we would be happy to explore it in future research.
>
> We sincerely hope that our response addresses your concerns and will be taken into consideration in your final evaluation. Thank you again for your time and effort!

---

### Official Review · Reviewer_zQ8b · 2025-06-30

**Clarity:** 3
**Significance:** 3
**Originality:** 2
**Rating:** 4
**Confidence:** 5

**Summary:**

The paper addresses the efficiency challenges of long-sequence inference in large language models (LLMs) by focusing on Key-Value (KV) cache eviction to reduce memory and I/O latency without sacrificing generation quality. It identifies a critical limitation in existing Top-k eviction methods: uniform allocation of a fixed cache budget across attention heads ignores the diverse concentration patterns exhibited by different heads. To overcome this, the authors propose Ada-KV, a head-wise adaptive budget allocation strategy guided by a theoretical upper bound on L1 eviction loss, which dynamically reallocates budget from sparse to dispersed heads. Ada-KV is designed to be plug-and-play, seamlessly integrating with prior SOTA methods such as SnapKV and Pyramid to form Ada-SnapKV and Ada-Pyramid variants. Extensive experiments on 13 Ruler datasets and 16 LongBench datasets, under both question-aware and question-agnostic scenarios, demonstrate that Ada-KV substantially improves post-eviction generation quality across various budget settings and base models (Llama-3.1-8B-Instruct and Mistral-7B-instruct-v0.2). Additionally, the paper presents an efficient CUDA implementation with a flattened cache layout and GQA compatibility, maintaining computational efficiency comparable to uniform eviction methods.

**Questions:**

1. Since Ada‑KV allocates a different number of tokens to each head, CUDA kernels will need to handle heterogeneous cache sizes per head. This variability may incur additional scheduling or memory fragmentation overhead, especially when batch‑processing multi-head operations. Could the authors clarify how this challenge is addressed in practice?

2. The Top‑k eviction step requires gathering token entries from non-contiguous memory locations, which breaks coalesced memory access patterns. Such gather operations are known to reduce memory bandwidth efficiency and increase latency. Could the authors clarify how this challenge is addressed in practice?

**Ethical Concerns:**

["NO or VERY MINOR ethics concerns only"]

**Final Justification:**

The score I have provided accurately reflects my current positive evaluation of the paper.

Especially, there are two "5" scores. I don't think this paper deserves three "5" scores considering that he main idea of this work remains relatively straightforward and incremental.

**Limitations:**

See weakness and questions.

**Quality:**

4

**Strengths And Weaknesses:**

# Strengths
1. Introduces the head-wise adaptive budget allocation framework for KV cache eviction, moving beyond uniform budget schemes.

2. Demonstrates seamless integration of Ada-KV with existing Top-k eviction methods (SnapKV and Pyramid), creating Ada-SnapKV and Ada-Pyramid with minimal changes to upstream frameworks.

3. Evaluates on two major benchmarks (Ruler and LongBench), across two base LLMs and both question-aware and question-agnostic settings, underscoring the broad applicability of the method.

4. Reports significant gains even under constrained budgets (e.g., 20% cache size), illustrating robustness in low-resource scenarios.

5. Proposes a flattened cache storage layout and custom CUDA kernels for variable-length FlashAttention, ensuring that adaptive allocation does not incur additional runtime or memory overhead compared to uniform methods.

6. Addresses Group Query Attention (GQA) compatibility, reducing redundancy in grouped KV caches and enabling a 4× cache size reduction in practice.

7. Offers detailed methodological descriptions, pseudocode (Algorithms 1 and 2), facilitating understanding and reproducibility.

# Weaknesses

1. The main idea of this work remains relatively straightforward. As such, the contribution may be perceived as a modest advancement over existing techniques

2. The theoretical analysis, while clearly structured, lacks sufficient depth and insight to represent a primary contribution of the work. Given its relatively simplicity, it would be more appropriate to condense this section into one or two succinct paragraphs, ensuring the paper remains focused on more impactful elements.

3. The safeguard parameter $\alpha$ (defaulted to 0.2) is introduced to prevent excessively small budgets for sparse heads, but there is limited analysis of its sensitivity or optimal selection across benchmarks.

---

> ### Author Rebuttal · Authors · 2025-07-31
>
> # Reviewer zQ8b
>
> Thank you for your thorough review. We address your questions as follows.
>
>
> ## Reply to W1:
>
> > The main idea of this work remains relatively straightforward. As such, the contribution may be perceived as a modest advancement over existing techniques
>
>
> The core strengths of Ada-KV lie in its simplicity, generality, and plug-and-play design. It has the potential to enhance a wide range of existing and future cache eviction methods without requiring additional training or model changes. Moreover, achieving effective head-wise budget allocation demands both algorithmic insight and efficient implementation—a non-trivial task when aiming to maintain high performance and scalability.
>
>
> ## Reply to W2:
>
> > The theoretical analysis, while clearly structured, .. Given its relatively simplicity, it would be more appropriate to condense this section into one or two succinct paragraphs...
>
>
> We appreciate your suggestion and will revise Section 3.2 to be more concise, so as to reallocate space to present more experimental results in the main text. While the theoretical analysis is simple by design, it provides a formal justification for head-wise allocation via a loss upper bound and improves interpretability. We will refine the section accordingly.
>
>
>
> ## Reply to W3:
>
> > The safeguard parameter (defaulted to 0.2) is introduced to prevent excessively small budgets for sparse heads, but there is limited analysis of its sensitivity or optimal selection across benchmarks.
>
> Due to the high cost of evaluating LLMs on long-text benchmarks, we conducted only small-scale sensitivity analysis for $\alpha$ setting in the initial stage and simply chose a fixed value rather than seeking the optimal setting for different models or budgets, as this would introduce significant complexity.
>
> Our results show that Ada-KV is generally robust to the choice of $\alpha$. A smaller $\alpha$ allows for more aggressive budget allocation, improving performance under limited budgets, while a larger $\alpha$ performs slightly better with higher budgets. Based on this trade-off, we set $\alpha = 0.2$ as a balanced result.
>
>
> | Longbench, Mistral-7B | Budget 128 | Budget 256 | Budget 512 | Budget 1024 |
> |---|---|---|---|---|
> |Ada-SnapKV $\alpha=$0 | **35.69** | **38.9** | 40.53 | 41.44 |
> |Ada-SnapKV $\alpha=$0.2 | 35.48 | 38.61 | 40.58 | 41.61 |
> |Ada-SnapKV $\alpha=$0.5 | 35.2 | 38.46 | **40.66** | **41.62** |
>
>
>
> ## Reply to Q1:
>
> > Since Ada‑KV allocates a different number of tokens to each head, CUDA kernels will need to handle heterogeneous cache sizes per head. This variability may incur additional scheduling or memory fragmentation overhead, especially when batch‑processing multi-head operations. Could the authors clarify how this challenge is addressed in practice?
>
>
> Firstly, the Top-K cache eviction method selects the most important subset from the original KV cache, copies them to new contiguous memory, and releases the original space. However, head-wise budget allocation does introduce management challenges.
>
> We address this challenge by implementing a CUDA kernel that flattens the cache of each head into a contiguous layout, along with metadata indicating the start position and length for each head. For batch processing, we use a variable-length flash attention technique that uses the metadata to extract relevant KV entries per head. Computation proceeds in a block-wise fashion along the sequence dimension, accomodating varying KV lengths across heads. Once computation for a shorter head is completed, GPU resources are dynamically reallocated to process blocks from other heads.
>
> Regarding memory fragmentation, the flattened layout ensures each head’s cache is contiguous in memory, so Ada-KV is no different from other cache eviction methods in this respect. Additionally, Ada-KV can further reduce memory fragmentation by supporting vLLM paged-attention. In accordance with the double-blind review policy, the code link will be provided after publication.
>
> ## Reply to Q2:
>
> > The Top‑k eviction step requires gathering token entries from non-contiguous memory locations, which breaks coalesced memory access patterns. Such gather operations are known to reduce memory bandwidth efficiency and increase latency. Could the authors clarify how this challenge is addressed in practice?
>
> The Top-K cache eviction method selects the most important cache entries from a large set, copying them into a new contiguous memory space, and releasing the original space. This ensures that the retained entries remain contiguous in memory after eviction.
> In practice, eviction occurs only once per layer during the prefill phase, meaning the operation is infrequent.
> So, the overhead of a single gather operation is negligible compared to the inference cost and the benefits it provides.

---

> > ### Comment · Reviewer_zQ8b · 2025-08-04
> >
> > Thank you again to the authors for their rebuttal. The score I have provided accurately reflects my current evaluation of the paper.

---

### Official Review · Reviewer_QpJ1 · 2025-06-30

**Clarity:** 3
**Significance:** 3
**Originality:** 3
**Rating:** 5
**Confidence:** 4

**Summary:**

The paper introduces Ada-KV, a novel method for optimizing Key-Value (KV) cache eviction in Large Language Models (LLMs) by employing adaptive head-wise budget allocation. Unlike prior cache compression methods that assign a uniform memory budget to each attention head, Ada-KV dynamically allocates budget based on attention concentration patterns. The paper establishes a theoretical framework showing that minimizing eviction loss can be guided by allocating more budget to heads with dispersed attention. This strategy is implemented in a plug-and-play fashion, improving existing methods like SnapKV and PyramidKV. Extensive experiments across datasets from Ruler and LongBench benchmarks, and multiple models (Llama-3.1-8B, Mistral-7B, Llama-3.1-70B), demonstrate significant improvements in generation quality, especially under low memory budgets and in the challenging question-agnostic setting.

**Questions:**

1. Have you explored applying Ada-KV during the decoding phase? What challenges or considerations arise there?
2. Regarding the safeguard, what would happen if you don't have safeguard, or have a high or low alpha?

**Ethical Concerns:**

["NO or VERY MINOR ethics concerns only"]

**Limitations:**

Yes

**Quality:**

3

**Strengths And Weaknesses:**

Strength:
1. This is the first paper to propose head-level KV cache budget allocation, addressing a clear limitation in prior work where all heads share the same fixed budget. The authors provide both theoretical motivation and empirical justification, showing that heads exhibit varied attention patterns, making uniform allocation suboptimal.
2. Ada-KV can be seamlessly integrated into existing eviction frameworks like SnapKV and PyramidKV. This modular design makes the method broadly applicable and easy to adopt without retraining or significant architectural changes.
3. The experiments is comprehensive, they do the experiments on multiple datasets with multiple model families. The method consistently outperforms baseline techniques across diverse datasets and models.
4. Ada-KV extends to models that use GQA, avoiding unnecessary cache duplication.

Weakness:
1. The method is only applied during the prefilling stage, meaning it does not affect the autoregressive decoding steps where cache size also grows with generation. While this is not a fundamental flaw, a discussion on extending adaptive budgeting to the generation phase would be a valuable addition.
2. While the safeguard mechanism for avoiding under-allocation to sparse heads is intuitively reasonable, its impact is not fully studied. Some ablation or sensitivity analysis on α would help understand the trade-offs.

---

> ### Author Rebuttal · Authors · 2025-07-31
>
> # Reviewer QpJ1
> Thank you for your valuable review and for acknowledging our work. We address your questions as follows.
>
> ## Reply to W1 & Q1:
> > ... a discussion on extending adaptive budgeting to the generation phase would be a valuable addition.
>
> Thank you for your suggestion. Like prior SOTA methods such as SnapKV and PyramidKV, Ada-KV currently applies cache compression only during the prefilling stage. That said, extending adaptive budgeting to the decoding phase is an interesting direction and certainly feasible.
>
> One possible approach is to segment the generated context into chunks and apply adaptive budgeting on these chunks (e.g., after decoding each chunk). We agree that it is a valuable direction for further work and plan to investigate compression strategies for the decoding phase.
>
>
> ## Reply to W2 & Q2:
> > While the safeguard mechanism for avoiding under-allocation to sparse heads is intuitively reasonable, its impact is not fully studied. Some ablation or sensitivity analysis on $\alpha$ would help understand the trade-offs.
>
>
> Thanks for your insightful question. Due to the high cost of evaluating LLMs on long-text benchmarks, we conducted only small-scale analyses for $\alpha$ setting in the initial stage and simply chose a fixed value rather than seeking the optimal setting for different models or budgets, as this would introduce significant complexity.
>
> Our results show that Ada-KV is generally robust to the choice of $\alpha$. A smaller $\alpha$ allows for more aggressive budget allocation, improving performance under limited budgets, while a larger $\alpha$ performs slightly better with higher budgets. Based on this trade-off, we set $\alpha = 0.2$ as a balanced result.
>
>
> | Longbench, Mistral-7B | Budget 128 | Budget 256 | Budget 512 | Budget 1024 |
> |---|---|---|---|---|
> |Ada-SnapKV $\alpha=$0 | **35.69** | **38.9** | 40.53 | 41.44 |
> |Ada-SnapKV $\alpha=$0.2 | 35.48 | 38.61 | 40.58 | 41.61 |
> |Ada-SnapKV $\alpha=$0.5 | 35.2 | 38.46 | **40.66** | **41.62** |

---

### Official Review · Reviewer_8Yky · 2025-07-02

**Clarity:** 4
**Significance:** 3
**Originality:** 3
**Rating:** 5
**Confidence:** 4

**Summary:**

This paper makes the observation that different attention heads within a layer have highly differing concentrations of attention scores across tokens in the KV cache. The authors first formulate the L1 Eviction Loss, prove that it is upper-bounded w.r.t. attention scores and eviction decisions, and show that KV cache eviction methods, and prove that the Top-$k$ eviction decision achieves the tightest upper-bound.

The authors then propose Ada-KV, a budget allocation algorithm adaptive to the token concentrations of attention heads, and show that this method is optimal w.r.t. the L1 Eviction Loss. The method, implemented with a custom CUDA kernel, is integrated as a subroutine and improves upon the strong baselines of SnapKV and PyramidKV for a number of long-context benchmarks.

**Questions:**

Can the authors please clarify the interaction of Ada-KV's "budget decision" with the actual eviction policy of the respective baselines SnapKV and PyramidKV in Algorithm 2?

I would appreciate it if the authors could address the questions raised in the "Weaknesses" section above.

**Ethical Concerns:**

["NO or VERY MINOR ethics concerns only"]

**Final Justification:**

My confusion surrounding the novel within-layer allocation of Ada-KV versus the cross-layer allocation of Pyramid-KV has been addressed, and has led to me appreciating the impact of Ada-KV over simply Snap-KV or simply Pyramid-KV. The paper appears strong to me, and contains a theoretical motivation, substantial empirical gains, and an efficient implementation.

**Limitations:**

The authors discuss the limitations (e.g. that the budget allocation is within-layer and not cross-layer) in Appendix A.5.

**Quality:**

3

**Strengths And Weaknesses:**

### Strengths

- The per-head allocation of the total token budget is formulated intuitively, and the simple Top-$k$ solution is shown to be a minimizer of the upper bound within the L1 Eviction Loss formulation. The approach is also strongly supported by observations of the behavior and visualization of attention heads.

- The CUDA kernel for Ada-KV retains the benefits of FlashAttention even when different heads have to attend to a variable number of tokens.

### Weaknesses

- While the L1 loss formulation appears to simplify the derivation of Ada-KV as minimizer for cache eviction loss, I do not see the intuition behind this theoretical formulation, i.e. that minimizing the Manhattan distance between attention scores is more useful than Euclidean distance. Additionally, the statement "[t]he degradation in generation quality after cache eviction stems from changes in the attention output" is trivially true but does not yield much insight into this theoretical formulation — especially since it's possible to construct larger attention score perturbations (w.r.t. L1 or L2 distance) which would actually result in better quality.

- It is not entirely clear to me to what extent the budget allocation of Ada-KV, which is directly responsible for which tokens get evicted, overrides the PyramidKV and SnapKV cache eviction strategies in Algorithm 2. While the role of Algorithm 1 is evident inside Algorithm 2, it is hard to understand the role of either of the two strategies. For example, PyramidKV allocates a smaller and smaller token budget at higher and higher layers. What if Ada-KV produces an "inverted pyramid" budget, and choose to "safeguard" more and more tokens at higher and higher layers? What would be the precise interaction of this method with PyramidKV in such a case?

- My previous point of confusion about the role of PyramidKV and SnapKV seems to be reflected in the experimental results, since the red and blue curves are nearly identical on the majority of datasets. This suggests to me that Ada-KV might be more of a KV cache eviction method in its own right than a "plug-and-play" integration with other methods.

---

> ### Author Rebuttal · Authors · 2025-07-31
>
> # Reviewer 8Yky
> Thank you for your recognition of our work and your insightful comments. We address your questions as follows.
>
> ## Reply to W1:
>
> > While the L1 loss formulation appears to simplify the derivation ... Additionally, the statement ... is trivially true but does not yield much insight into this theoretical formulation — especially since it's possible to construct larger attention score perturbations (w.r.t. L1 or L2 distance) which would actually result in better quality.
>
> We chose L1 distance primarily for its simplicity. That said, we agree that other metrics—such as L2 or more task-specific distances—may better capture attention perturbations in certain contexts. Exploring these alternatives is a valuable direction for future work.
>
> While it is true that larger perturbations can occasionally result in better generation quality, our theoretical motivation relies on the commonly assumed Lipschitz continuity of neural networks. Under this assumption, smaller input perturbations generally lead to smaller output variations. Therefore, reducing attention output perturbations after cache eviction decreases variation in the subsequent FFN block and deeper layers, ultimately narrowing the gap between the token distributions of the original and post-eviction outputs. This keeps generation after cache eviction as close as possible to the original, which is the primary goal of compression quality. We will clarify this intuition in the theoretical section.
>
>
> ## Reply to W2 & Q1:
>
> > ...What if Ada-KV produces an "inverted pyramid" budget, and choose to "safeguard" more and more tokens at higher and higher layers? What would be the precise interaction of this method with PyramidKV in such a case?...
>
> Sorry for the confusion. In fact, Ada-KV performs dynamic allocation among all attention heads **within a layer**, and thus does not conflict with previous methods that handle allocation **across layers**.
>
> SnapKV uses a uniform token budget for all layers, while PyramidKV sets different layer-wise budgets. However, both methods distribute each layer’s budget evenly across all heads within that layer, without further head-wise differentiation, mostly due to the complexity of dynamic allocation and parallelization at that granularity.
> Ada-KV addresses this by dynamically allocating each layer's budget among the attention heads within this layer, enabled by our theoretical formulation and an efficient CUDA kernel that facilitates the implementation.
>
>
> Thus, Ada-KV and SnapKV/PyramidKV serve orthogonal roles:
> - SnapKV/PyramidKV define how many tokens to retain per layer.
> - Ada-KV decides how to split that per-layer budget among the heads.
>
> These mechanisms do not conflict or override one another. In future work, we plan to extend Ada-KV's adaptive allocation strategy to layer-wise budget distribution, which may bring further improvements.
>
> ## Reply to W3:
>
> > ... since the red and blue curves are nearly identical on the majority of datasets. This suggests to me that Ada-KV might be more of a KV cache eviction method in its own right than a "plug-and-play" integration .
>
> Thank you for your question! The red and blue curves are very close because the previous two SOTA methods SnapKV and PyramidKV have similar performance, which leads to a small performance gap between Ada-SnapKV and Ada-PyramidKV. This reflects the similarity of the underlying baselines, rather than a limitation of Ada-KV.
>
> Importantly, Ada-KV remains a plug-and-play enhancement: it dynamically allocates budget within layers and can be applied to base cache eviction method.
> Several new methods were released publicly after our submission (to maintain double-blind review, we refer to them as Anonymous Methods; links will be provided after publication). We found that Ada-KV could substantially improve their performance. We will include these new results as case studies in the Appendix of the revised version after publication.
>
>
>
> | Question-Agnostic LongBench Ave. Score | 20% Cache | 40%  Cache |
> |---|---|---|
> | SnapKV  | 40.21 | 44.92 |
> | SnapKV + Ada-KV (Ada-SnapKV) | 41.39 | 45.74 |
> | Anonymous Method | 43.78 | 47.76 |
> | **Anonymous Method + Ada-KV** | **46.68** | **49.21** |

---

> > ### Comment · Reviewer_8Yky · 2025-08-04
> >
> > Thank you to the authors for their response.
> >
> > I apologize for my confusion on W2 — my original review conflated the novelty of the adaptive *within-layer* budget allocation among heads provided by Ada-KV with the *cross-layer* budget offered by Pyramid-KV. The resolution of W2 has led to a resolution of W3.
> >
> > I am raising my scores accordingly.

---

### Note · Authors · 2025-08-12

Dear Area Chair,

We thank all reviewers for their feedback and  Reviewers 8Yky, QpJ1, and zQ8b for their acceptance consensus. They affirmed our work’s novelty, efficient implementation and broad applicability as a plug-and-play enhancement.

While we appreciate Reviewer 1zqk acknowledging our work as **"an inspiring contribution to cache eviction field"** , their rejection is rooted in **a fundamental misunderstanding of the whole cache eviction field itself**—an inappropriate basis for judging our work. Their argument, that cache eviction is obsolete compared to dynamic sparsity field, is flawed because:

## It Conflates Orthogonal Research Fields.
- Cache Eviction Field (Our Work): **Compresses the cache's memory footprint** by evicting unimportant KV entries.
- Dynamic Sparsity Field: Retains the full KV cache (often offloaded) and **accelerate computation by accessing only a small subset of tokens per decoding step**.

These fields are not competing but synergistic. One can first apply cache eviction method to compress the cache size, and then use dynamic sparsity to further accelerate computation on that smaller cache. Progress in one field does not invalidate the other.

## Its  "budget" comparison is misleading as it equates two different concepts.
- Cache Eviction Budget (e.g. 20%): the percentage of cache **stored** in memory.
- Dynamic Sparsity Budget (e.g. 5%) : the percentage of cache **computationally accessed per decoding step** (while full cache remains stored) .

To illustrate this flawed logic, let's consider another complementary technique: KV cache quantization. Cache quantization from BF16 to INT8 or INT4 reduces the cache memory to 50% or 25%, respectively, much like cache eviction does. **Following the Reviewer 1zqk's logic, one would have to conclude that dynamic sparsity's smaller "5% budget" would also make KV cache quantization field obsolete.** This is patently false, as their respective "budgets" refer to entirely different concepts, and the techniques themselves are, in fact, complementary.

Currently, the cache eviction field remains active and important, and Ada-KV can enhance many continually emerging methods. **We believe it is inappropriate to dismiss this actively progressing field based on a flawed comparison, and subsequently use that as grounds for rejecting our work.**

We hope that this clarification has been helpful in resolving the confusion. Thank you for your consideration.

Best regards,
The Authors

---

### Decision · Program_Chairs · 2025-09-17

**Decision:**

Accept (poster)

**Comment:**

Ada-KV introduces the first head-wise adaptive KV cache allocation strategy, optimizing LLM inference. Its strengths include plug-and-play integration without retraining, significantly improving generation quality under low memory budgets. Reviewers initially questioned the L1 loss intuition, interaction with baselines, and safeguard parameter sensitivity. Authors clarified L1 was for simplicity and explained Ada-KV's orthogonal within-layer/cross-layer interaction with existing methods. Crucially, the core disagreement regarding the conflation of KV cache eviction with dynamic sparsity was refuted, emphasizing these are orthogonal and active research fields. The paper was accepted due to its novelty, effectiveness, and clarified contribution to the field.